# An indicator of sea ice variability for the Antarctic marginal ice zone

Marcello Vichi[1,2]

[1]Department of Oceanography, University of Cape Town, 7701, Rondebosch, South Africa
[2]Marine and Antarctic Research centre for Innovation and Sustainability (MARIS), University of Cape Town, 7701, Rondebosch, South Africa

**Correspondence:** Marcello Vichi (marcello.vichi@uct.ac.za)

**Abstract.** Remote-sensing records over the last 40 years have revealed a large year-to-year global and regional variability in Antarctic sea ice extent. Sea ice area and extent are useful climatic indicators of large scale variability, but they do not allow to quantify regions of distinct variability in sea ice concentration (SIC). This is particularly relevant in the marginal ice zone (MIZ), which is a transitional region between the open ocean and pack ice, where the exchanges between ocean, sea ice and atmosphere are more intense. The MIZ is circumpolar and broader in the Antarctic than in the Arctic. Its extent is inferred from satellite-derived SIC using the 15-80% range, assumed to be indicative of open drift or partly closed sea ice conditions typical of the ice edge. This proxy has been proven effective in the Arctic, where there is a good correspondence between sea ice type and sea ice cover. It is less reliable in the Southern Ocean, where sea ice type is less linked to the concentration value, since wave penetration and free drift conditions have been reported with 100% cover. The aim of this paper is to propose an alternative indicator for detecting MIZ conditions in Antarctic sea ice, which can be used to quantify variability at the climatological scale on the ice-covered Southern Ocean over the seasons, as well as to derive maps of probability to encounter a certain degree of variability in the expected monthly SIC value. The proposed indicator is based on statistical properties of the SIC; it is derived from the standard deviation of daily SIC anomalies with respect to the monthly 40-years climatology, a method often employed in the climate sciences. The use of a monthly climatological mean as the baseline allows to capture changes due to both the seasonal advancement/retreat and the local weather-driven variability typical of less consolidated sea ice conditions. This method has been tested on the available climate data records to derive maps of the MIZ distribution over the year, and compared with the threshold-based MIZ definition. It maintains the same regional clustering observed in the standard diagnostics, but also allows to quantify the mean intensity of MIZ variability in these regions, which can be used to assess the skills of sea ice models in forecasting or climate modes. It offers a revised view of the circumpolar MIZ seasonal cycle, with a rapid increase of the extent and a saturation in winter, as opposed to the steady increase from summer to spring reported in the literature. It also reconciles the discordant MIZ extent estimates using the SIC threshold from different algorithms. This indicator complements the use of the MIZ extent and fraction, allowing to derive the climatological probability of exceeding a certain threshold of SIC variability, which can be used for planning observational networks and navigation routes, as well as detecting changes in the variability when using climatological baselines for different periods.

## 1 Introduction

The Southern Ocean holds the largest circumpolar marginal ice zone (MIZ) in the global ocean (Weeks, 2010, p. 408), while the Arctic MIZ regions are mostly confined to the Bering Sea and the Greenland and Norwegian Seas (Wadhams, 2000). In most general terms, and independently of the hemisphere, the MIZ can be depicted as a band of young or fractured ice with floes smaller than a few hundred metres, which is continuously affected by air-sea interactions in the form of heat exchanges, wind and current drag, and wave action (Häkkinen, 1986; Dumont et al., 2011; Williams et al., 2013; Zippel and Thomson, 2016; Sutherland and Dumont, 2018; Squire, 2020).

### 1.1 Definitions of the MIZ: sea ice concentration, wave penetration and ice type

The MIZ is a transitional region, and as such, it is often defined by contrasting consolidated pack ice against open ocean conditions. This implies the identification of two boundaries, one at the ice-ocean margin and one within the pack ice. The ocean edge and the MIZ extent are inextricably linked, since it is difficult to find sharp separations between these two components. Hence, the definition of the MIZ in the literature depends on the properties that are of interest in each study, and often on the polar hemisphere considered. Following on from Arctic studies, the boundaries are derived from contour lines of sea ice concentration (SIC), the fraction of ice-covered water obtained through passive microwaves sensors onboard satellites (Comiso and Zwally, 1984; Meier and Stroeve, 2008/ed; Strong and Rigor, 2013; Stroeve et al., 2016). Operationally, the MIZ is defined as that region of the sea ice where SIC is comprised between 15 and 80%, and the MIZ extent depends on how the distance between these contours are computed (Strong et al., 2017). This definition is tightly linked to the SIC retrieval from satellites, since the limit of 15% is considered to be a viable rule-of-thumb to overcome the uncertainties in the methodology (Comiso and Zwally, 1984). Within this range, sea ice is assumed to be in open pack conditions, with higher chances of drifting ice and the penetration of gravity waves due to the floes being smaller than the wave length (Squire, 2020). The threshold-based MIZ definition has been directly applied to Antarctic sea ice despite the remarkable differences in sea ice formation processes (e.g. Weeks and Ackley, 1986; Petrich and Eicken, 2017; Maksym, 2019). As an alternative definition, it has been proposed to estimate the MIZ extent based on the region where the wave field is responsible for setting the sea ice thickness (Williams et al., 2013; Sutherland and Dumont, 2018). Rolph et al. (2020) argue that, even if the use of more physical concepts such as the penetration of waves is a valid definition for studies of the MIZ, comparisons of MIZ extent between model and observational products should be based on SIC thresholds. The analysis of the MIZ fraction of the total cover based on SIC thresholds has shown promising results to benchmark the skill of climate models and their response to atmospheric warming (Horvat, 2021). Sea ice in the MIZ is therefore of a special kind, which responds differently than pack ice to the environmental drivers and may have relevant climatic implications, at least in the Arctic.

However, the relationship between SIC, ice type and ice properties is not yet constrained in the Southern Hemisphere. Ice type is still an ambiguous term in the literature, because it is used differently in different contexts. In predominantly seasonal sea ice as found in the Antarctic, with continuous transition between new and young ice and the dominance of frazil ice (Matsumura and Ohshima, 2015; Haumann et al., 2020; Paul et al., 2021), the exchanges of energy across the interface may

be less dependent on the degree of coverage, and rather be more affected by the composite of the sea ice texture. Ice type is derived from direct observations, using categories like the WMO nomenclature and codes (WMO, 2014, 2021), and the

SCAR Expert Group on Antarctic Sea-ice Processes and Climate, ASPeCt (https://www.scar.org/science/aspect/home/). These classified features of sea ice heterogeneity do not necessarily co-vary with SIC or thickness, which means that young ice of less than 30 cm thick with a combination of pancake and frazil ice can still have 100% cover (Figure 1c), which is susceptible to wave penetration. Wave attenuation is considered to be a function of ice type (ice properties), which is ultimately approximated to sea ice concentration (Mosig et al., 2015; Squire, 2020) for lack of better assumptions. This creates circular reasoning, since

we are looking to define the MIZ extent based on waves that depend on ice properties that we cannot measure, and hence we resort to the observable variables: SIC, mean wave period, and wind direction. Based on recent observations in the Ross Sea in autumn, Montiel et al. (2022) have found that simple parameterizations of attenuation are unlikely to capture the wide range of sea ice conditions found in the Southern Ocean.

It is no surprise that the SIC-based definition of the MIZ is thus the one most often used to estimate temporal trends in

the MIZ extent at both poles (Strong and Rigor, 2013; Strong, 2012; Stroeve et al., 2016; Rolph et al., 2020; Horvat, 2021), with contrasting results that may be partly attributed to methodological issues (Strong et al., 2017). Stroeve et al. (2016) found large differences in estimating the seasonal cycle of the Antarctic MIZ extent using different algorithms. Over a climatological seasonal cycle, the Bootstrap method returned a higher percentage of consolidated pack ice than the NASA Team algorithm, and this led to differences in the trend analyses.

## 1.2 Characterizing variability in Antarctic sea ice

A most pressing question is not whether the MIZ has been increasing or decreasing in the Antarctic and how different it is from the Arctic, but rather if the Antarctic MIZ features and variability can be properly captured using the threshold-based concentration criteria. With variability I refer to the daily change in SIC over a monthly scale in a climatological sense, and I will expand later on the roles of spatial and temporal variability. In the Antarctic, the MIZ is a characteristic of the advancing

edge, since during this phase sea ice progresses northwards and expands zonally due to the increase in ocean surface towards the equator. This leads to divergence, and lowers the chances of rafting and ridging, which are still considered the main thickening mechanisms in the Southern Ocean ((Worby et al., 1996)). An analysis of one ice-tethered buoy deployed in the Eastern Antarctic sector revealed a large MIZ band of almost 300 km that persisted throughout the winter expansion until early December (Womack et al., 2022), with satellite-retrieved ice cover permanently above 100%. In this region there would be no

exchange through the ice between the ocean and the atmosphere, largely underestimating the possible fluxes. Almost all the proposed parameterizations of energy, momentum and gas exchange through sea ice are linearly dependent on the area cover fraction (Steele et al., 1989; Martinson and Wamser, 1990; Worby and Allison, 1991; Andreas et al., 1993; Martinson and Iannuzzi, 1998; Bigdeli et al., 2018; Castellani et al., 2018; Gupta et al., 2020).

Due to the lack of better observational constraints, changes in remotely sensed SIC over a range of time scales should still be

used as the main indicator of responses to the environmental drivers. In the following, I will refer to these features and drivers as the 'MIZ characteristics', even if they occur in areas that are distant from the sea ice edge. The atmospheric and oceanic drivers

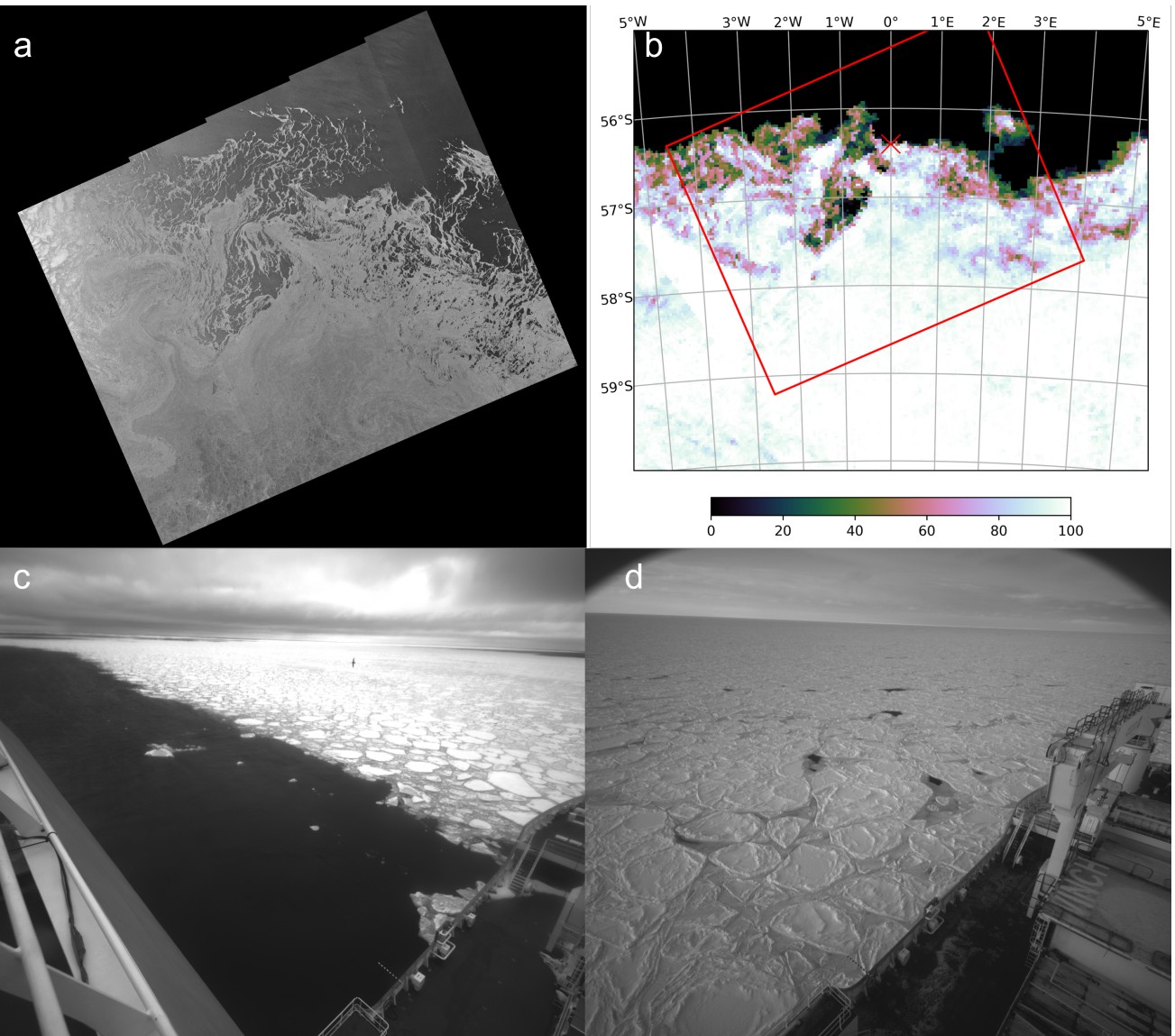

**Figure 1.** Example of sea ice conditions in the marginal ice zone at the edge with the ocean in austral spring and about 200 km into the pack ice in austral winter. (a) SAR image from the European Space Agency Sentinel 1B (GRD, acquired on 2019-10-21T19:21:53, obtained from http://www.seaice.dk at 300 m resolution, with credits to Roberto Saldo, dtu space and Technical University of Denmark). (b) Sea ice concentration from AMSR2 on the same day in stereographic polar projection (3.125 km resolution, processed by the University of Hamburg and obtained from ftp://ftp-projects.cen.uni-hamburg.de/seaice/AMSR2/3.125km) showing the footprint of the SAR image and the location of the icebreaker SA Agulhas II in the morning of 2019-10-22. (c) Sea ice conditions before entering the MIZ at the location shown by the red cross in panel b. The sharp transition is the wake of the ship after reaching the sampling position. (d) Cemented pancake ice floes in austral winter (observed from the SA Agulhas II on 2019-07-27 at 0E, 57S).

that are more active in these regions modify the sea ice area and extent, and should not be considered absent in regions with 100% cover. There is growing evidence in the Southern Ocean that: 1) extended regions with mixed types of sea ice in a fully covered ocean show MIZ characteristics from austral winter to spring (Alberello et al., 2019; Vichi et al., 2019; Alberello et al., 2020; Womack et al., 2022); 2) waves penetrate deep into the pack ice throughout the seasons (Kohout et al., 2014, 2015; Stopa et al., 2018; Massom et al., 2018; Kohout et al., 2020), and 3) extended regions of high variability in sea ice concentration and drift can be found in correspondence with large scale synoptic events (Vichi et al., 2019; de Jager and Vichi, 2022). Figure 1 gives an example of the complex conditions observed in the Antarctic MIZ. The Synthetic Aperture Radar (SAR) image shows a pattern of the MIZ in austral spring that is very well captured in the AMSR2 data, which reports 100% concentration very close to the edge where the ship was located (Fig. 1b). The ice-covered ocean was confirmed, but sea ice type was classified as grey-white ice, with a combination of thin fragments and frazil ice from refreezing (Fig. 1c). These conditions extended southward throughout the area of 100% cover. Also in regions of cemented pancakes as shown in Fig. 1d, waves associated to intense extratropical cyclones can penetrate and modify the surface features extensively (Vichi et al., 2019). Finally, we should also note the confounding effect of building composites from satellite swaths, with a clear discontinuity line in the SIC field between 3-5E in Fig. 1b, that is indicative of substantial sub-daily changes in the sea ice cover. A threshold-based indicator of MIZ characteristics may thus lead to erroneous definition of sea-ice characteristics and their parameterization in models, with unpredictable consequences on the design of observational campaigns and model predictions.

## 1.3 The need for a novel indicator

The growing body of observations poses the problem of a proper description of the Antarctic MIZ, and of Antarctic sea ice variability in general. Every latitude of the Southern Ocean, apart from the few regions of multi-year ice, can be classified as seasonal sea ice zone. This implies that for a period of time of variable duration, the sea ice may present MIZ characteristics, which may not necessarily be found at the margin of the ice-covered region. In this work I reassess the assumption that absolute thresholds of SIC contain sufficient information to characterize Antarctic sea ice, in contrast with the Arctic, where a better correspondence between ice cover fraction and ice type allows to discriminate first year (seasonal) ice from multi-year ice, with the subsequent emergence of categories based on thickness and ice-age. This is less relevant in the Antarctic, where the majority of sea ice is thin and seasonal. Antarctic sea ice and its MIZ features cannot thus be decomposed in further categories, unless through direct observations or the use of high-resolution SAR images, which are limited in space and time. Given that the only available data at the planetary scale are passive microwave data of brightness temperature, there is merit in investigating whether smaller changes in pixel concentration from remote sensing hold some consistent measure of change in the ice character.

In the following sections, I will demonstrate how the use of an indicator based on the SIC standard deviation of daily anomalies computed over the monthly time scales allows to reconcile the mismatch observed in the seasonality of the MIZ extent in the Antarctic when using different satellite products. This indicator is meant to quantify the temporal variability of SIC over each month, and I will compare its magnitude against the spatial variability, to show that time variability is an intrinsic feature of the MIZ. This variability combines together the advance/retreat of sea ice within a month, as well as the

daily changes in SIC caused by the passage of storms (e.g. Vichi et al., 2019). I will then investigate sub-seasonal scale variability in SIC with the aim to construct climatological maps of MIZ features in Antarctic sea ice, as a complementary information to the threshold-based classification. The interest here is not whether the retrieval of brightness temperature is measuring the actual concentration of ice-free versus ice-covered ocean, but rather if the relative time-change of this proxy is representative of a physical variation in sea ice state. In this first work, I will not link the observed variability to the possible drivers, but I will present the advantages of this method with respect to the threshold-based MIZ definition. Further analyses can be done eventually based on this rationale. In the following, the indicator will also be used to construct climatological maps of SIC variability and probability of exceeding extreme values of variability, hence assisting with long-term navigation planning, design of observational experiments and assessment of model outputs.

## 2   Methodology

### 2.1   Remote sensing data

The analysis was carried out using SIC data from the sea ice Climate Data Records (CDR) from NOAA/NSIDC (Peng et al., 2013; Meier et al., 2021, version 3 and 4) and from the EUMETSAT OSI SAF (OSI-450) product (Lavergne et al., 2019). The two datasets were initially chosen for their different approaches. The NOAA/NSIDC CDR until version 3 (Meier et al., 2017) represented a level 3 product that followed all the standards for traceability and reproducibility with minimal filtering; since version 4 it is now a level 4 product, with additional gap-filling procedures that have been introduced to make the estimates of sea ice extent (SIE) more comparable with other products (Windnagel et al., 2021). The OSI-450 product is a gap-less, level 4 product, which includes additional manual corrections and spatial/temporal interpolations to fill data gaps. The data processing of OSI-450 also used an open-water filter aimed at removing weather-induced false ice over open water, which may also remove some true low-concentration ice in the MIZ (Lavergne et al., 2019). OSI-450 provides a variable containing the raw data, which has been used to further assess sea ice variability.

The NOAA/NSIDC CDR product is meant to be an improvement on the individual algorithms, namely the NASA Team (NT) and the Bootstrap (BT). The rationale behind this choice is that passive microwave algorithms tend to underestimate concentration during the summer melt season (Meier et al., 2014). Since greater underestimation is typical in the BT algorithm, the CDR implements a 10% cut-off of this field and maximises the values between the two above the threshold. This means that all values lower than 10% from the BT product are not included in the CDR. As indicated in Sec. 1.1, the NT and BT algorithms have shown major differences when estimating the MIZ extent and its seasonality (Stroeve et al., 2016). The CDR will then be compared against the individual products because the rationale for its construction does have an impact on the MIZ estimation.

For the purpose of this analysis that focused on daily variability, the NOAA/NSIDC CDR version 3 was preferred for the lower level of smoothing and aliasing, which highlighted conspicuous features of the MIZ. With the new version 4 and likely the future versions, the NOAA/NSIDC CDR has implemented the spatial and temporal filtering, which were in version 3 only applied to the Goddard merged product, that extended the period back to January 1979. The NOAA/NSIDC CDR has

practically substituted the Goddard merged product, and it is more similar to the OSI-450 in terms of large scale properties. To reproduce the results observed in version 3 (not available on line anymore) the analysis has been performed on a reprocessed version, which corrects some bugs in version 4, removes the interpolated pixels and focuses on the period 1987-2019, for which daily data are mostly available. The scripts for this processing are available in the supplementary material. In the following, the results will be discussed against the other data sets and the corresponding figures for the NOAA/NSIDC CDR version 3 and OSI-450 CDR are available in the supplementary material.

## 2.2 Statistical analysis of variability

The methodology treats the variability of remote-sensed SIC as if it were a perturbation around an expected value. In the following, SIC is expressed as the fraction between 0 and 1; this value is assumed to be an objective measure of sea ice state rather than an actual indicator of ocean coverage. Regions of closed pack ice, or of ice-free ocean outside the seasonal ice zone are more likely to experience small variations around a long-term mean value of the SIC (close to 1 in the former case and to 0 in the latter). Persistent conditions of multi-year ice and permanently ice-free ocean will have less noise, hence a negligible dispersion around the climatological mean. The standard deviation of the daily SIC anomaly with respect to a chosen reference value can thus be used to measure the degree of variance in sea ice conditions experienced by a certain pixel over a month.

The daily SIC anomaly for each pixel is computed by subtracting the daily SIC from the monthly climatology $\overline{C}^n$:

$$a_i^m = C_i^m - \overline{C}^n, \tag{1}$$

where the index $i$ runs over the number of days in month $m$ and $n = 1, \ldots, 12$ indicates the month of the year. The index $m$ runs over the total number of months in the time series (e.g. 396 for NOAA/NSIDC CDR). The reason for choosing the monthly climatology as the reference value is crucial for the analysis and further explained below. Since the variable SIC is constrained between 0 and 1, so is the anomaly. The standard deviation of the daily anomaly is then computed for each month, to measure the spread around the climatological SIC monthly mean as follows:

$$\sigma_{SIA}^m = \sqrt{\frac{1}{N} \sum_{i=1}^{N} (a_i^m)^2}, \tag{2}$$

where $N$ is the total number of days in the month. The standard deviation is effectively a sum of squares, since the mean of the anomalies is null. The climatological monthly standard deviation of the anomalies ($\overline{\sigma}_{SIA}^n$) has also been computed by pooling together all the daily anomalies from the same month in different years

$$\bar{\sigma}_{SIA}^n = \sqrt{\frac{1}{N} \sum_{j=1}^{N \times Y} \left(a_j^n\right)^2}, \tag{3}$$

with $Y$ is the number of years, and the index $j$ runs over the number of days of the Januaries, Februaries, etc. The variable $\sigma_{SIA}^m$, hereinafter referred to as "the indicator" $\sigma_{SIA}$ with the index $m$ dropped, describes a left bounded distribution, where the value 0 indicates lack of SIC variability over the month and the maximum expected value is 0.5. The exclusion of the zeroes represents an unbiased distribution of SIC variability.

This analysis does not deliberately discriminate between a point that is experiencing a seasonal transition of the MIZ band during sea ice advance or a persistence of short-term variable SIC conditions more typically ascribed to the ice edge. This is the main reason for using the daily anomaly against the monthly climatology instead of the daily climatology (based on daily values or daily running means over a weekly to monthly time window). The use of a filtered background climatology with a window shorter than a month would include the smooth daily transition during the advance and retreat phase. It does retain some measure of variability but reduces the variance of the signal due to the meridional advancement, which is a fundamental characteristics of the MIZ. On the other hand, this same analysis conducted over the weekly scale would enhance the role of synoptic forcing. The method chosen here encompasses both aspects. Since the anomaly is computed roughly over the same number of days for each pixel (excluding the random missing data), it is more likely that a rapid transition between new, young and first-year ice would result in an overall lower value of the monthly indicator, nevertheless recording the information that this region of the ocean has been partly interested by changes in SIC.

The difference between the temporal variability expressed by this index and the spatial variability has been analysed by comparing with the NOAA/NSIDC CDR derived variable `stdev_of_cdr_seaice_conc`, which computes the spatial standard deviation of the box of 9 pixels surrounding each pixel. This measure takes into account the uncertainty of a SIC value based on the variability in the adjacent pixels. I used the monthly average of the latter, and I assumed that the $\sigma_{SIA}$ indicator is a valid measure of temporal variability indicative of MIZ conditions when the ratio with the spatial variability is smaller than 1.

The indicator is finally used to estimate the chances of encountering variable MIZ conditions at each pixel on a monthly climatological time scale. The probability of an ocean region being affected by MIZ conditions during a given month has been computed using the empirical exceedance (which is equivalent to 1 - CDF, the cumulative density function, when the function is known):

$$EP = 1 - \frac{r}{N} \tag{4}$$

where $r$ is the rank of the sorted series of $\sigma_{SIA}^m$ values. Given a certain threshold of the indicator that is known to correspond to MIZ conditions, this function gives an empirical estimate of the probability to exceed that value.

## 3 Results

### 3.1 An indicator of climatological variability for the MIZ

The empirical distribution of $\sigma_{SIA}$ follows a Pareto distribution (Fig. 2). In a Pareto distribution, the median is biased towards the lower values, indicating a majority of pixels with low SIC variability, but the tail of the distribution is sufficiently fat to have an influence. The cumulative density function is a power law, which can be fitted well with the Pareto function (p-value of the Kolmogorov-Smirnov test virtually zero; the test had to be run on sub-samples for computational reasons). The empirical function slightly departs from the fitted distribution for values above 0.1, which could be indicative of the superposition of two distributions.

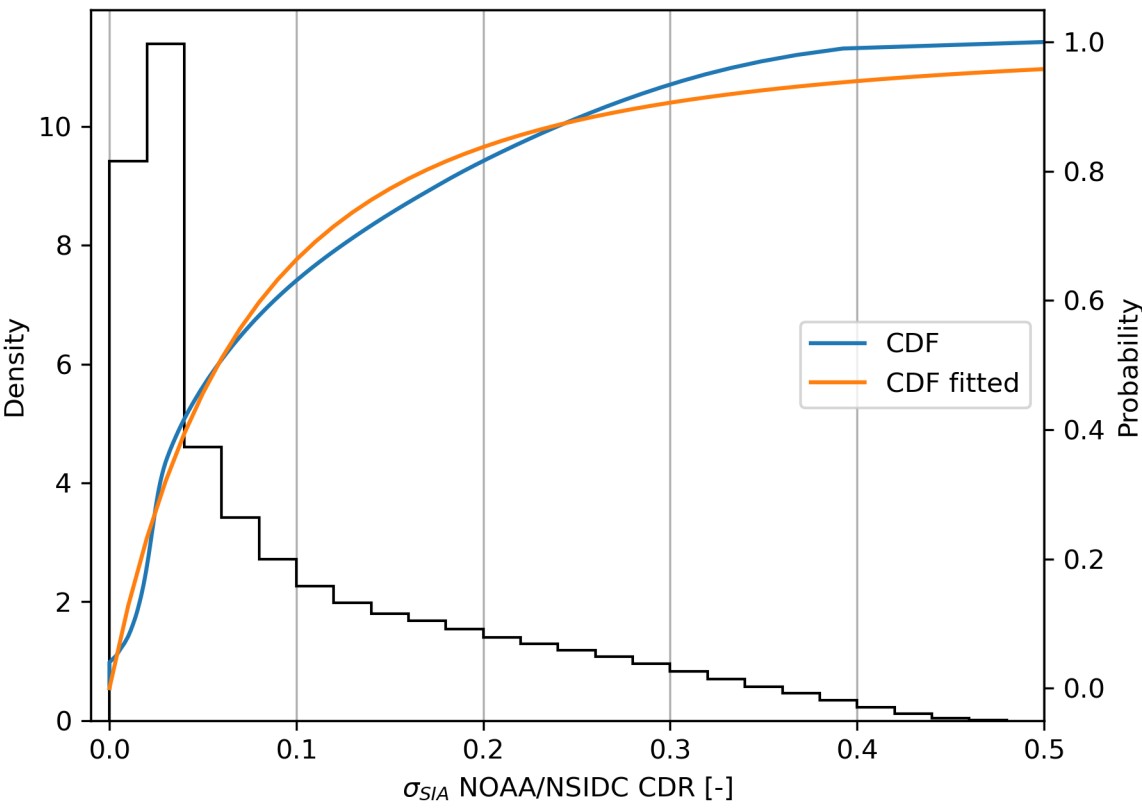

**Figure 2.** Empirical probability (black line) and cumulative (blue line) density functions of the $\sigma_{SIA}$ indicator from the NOAA/NSIDC CDR data set. The orange curve is the fitted Pareto distribution.

Since the interest is in identifying the typical conditions differentiating the MIZ pixels from those belonging to consolidated and less variable SIC regions, the median of the indicator computed for each pixel is a useful descriptor for obtaining a map of spatial features (Fig. 3a). Higher values of the $\sigma_{SIA}$ median are indicative of larger departures from the long-term conditions (when sea ice is present in the region). These highly variable regions are found in the outer part of the sea ice as expected. They
225    are distributed zonally in a rather homogeneous way, with a few peaks in the Bellingshausen, Eastern Weddell Sea (13°E) and Ross Sea (150°W) regions, located close to areas of interruptions of the zonal belt. Another area of high median is associated to coastal polynyas. These are known regions, in which the SIC is recognised to be more variable and usually less consolidated. A greater halo of scattered variability is observed mostly in the Atlantic and eastern Antarctic sectors, extending to about 55°S. This halo is removed when the analysis is run on the unprocessed CDR (see Sec. 2.1 for more details) and OSI-450, which are
230    gap-less and/or filtered, and it is enhanced in the NOAA/NSIDC CDR V3 (Fig. S1 and S2 in the supplementary material).

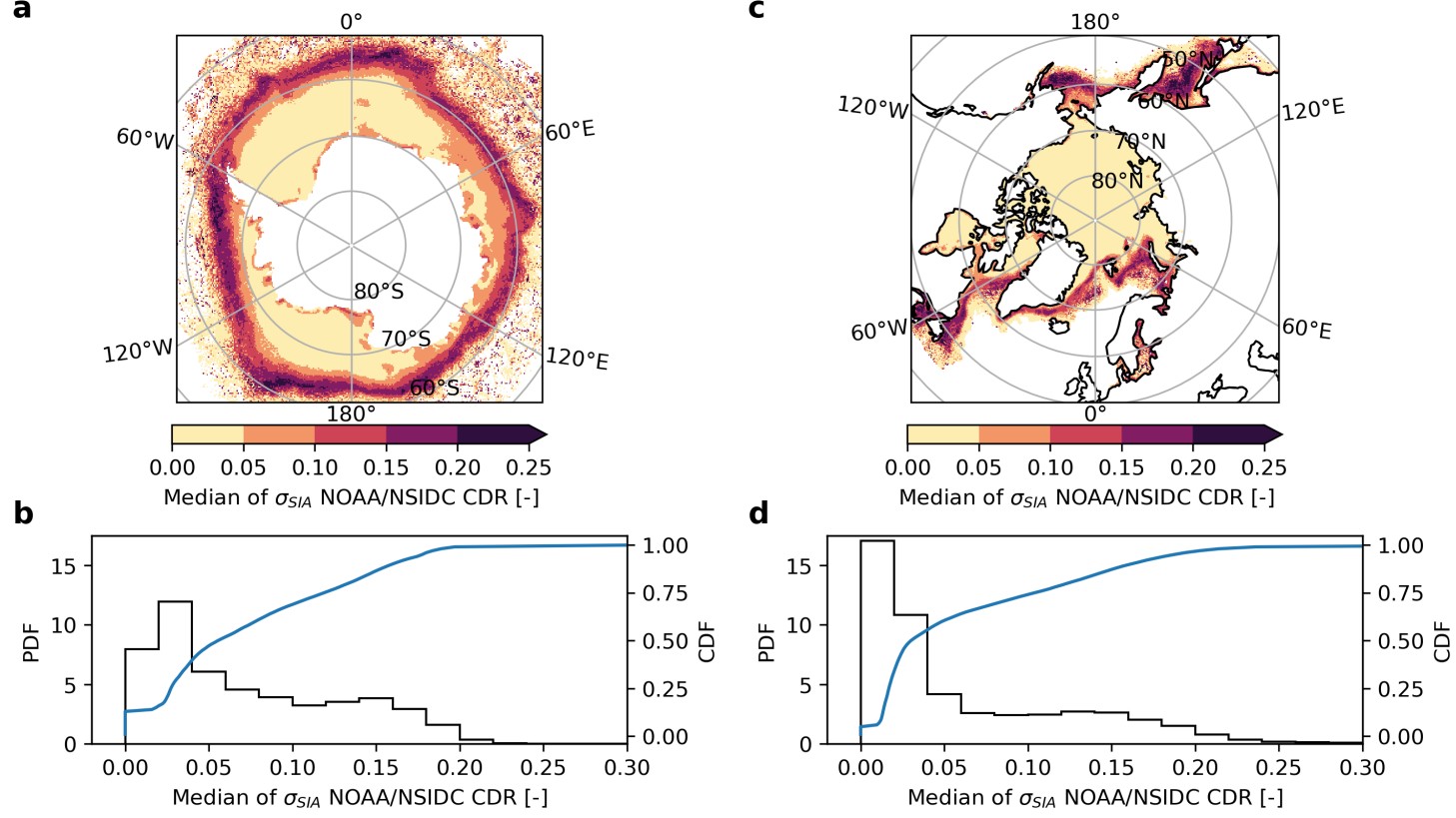

**Figure 3.** (a) Median of the $\sigma_{SIA}$ indicator on a stereographic projection. The pixels with SIC=0 and $\sigma_{SIA} = 0$ have been excluded from the analysis. (b) Empirical probability and cumulative density functions of the median values from the map shown in panel a (PDF: black line and CDF: blue line). (c-d) same as (a-b) but for the Arctic. All data are from the NOAA/NSIDC CDR (1987-2019).

The median distribution shown in Fig. 3b confirms the presence of different processes underlying the variability of Antarctic SIC. The distribution of the $\sigma_{SIA}$ median is more log-normal and bimodal than the overall sample distribution presented in Fig. 2, with maximum values below 0.3 (0.2 is the 99th percentile). There is still a large percentage of values with very low intraseasonal variability (which was not found in V3, see supplementary Fig. S2), but the bimodality is evident. The first peak

235 is larger and centred around 0.03 and the second one is above 0.15, with a trough between 0.1 and 0.15. The change of slope in the empirical CDF is more evident here, and corresponds to the range of values where the two distributions presumably intersect. By combining the spatial map with the distribution of the median, we can say that values between 0.1 and 0.15 indicate mixed regions were consolidated pack ice may show concentration changes akin to the features observed at the ice margin, and values above 0.15 can be clearly identified as having MIZ-like features.

240 The same analysis done for the Arctic (Fig. 3c-d) indicates that the regions of higher temporal variability of SIC at the sub-seasonal scale are narrow and confined to the Bering, Greenland, Irminger, and Norwegian Seas areas, as reported in the

literature (Wadhams, 2000). The empirical distribution of the median is also different from the Antarctic. The number of pixels with low variability is larger, as known to be in the Arctic due to the presence of multi-year ice, and the second peak is lower and barely visible. There is instead a plateau of points that show median values of the indicator between 0.05 and 0.17, and a clear threshold is less distinguishable.

In the following, I will only focus on Antarctic sea ice, and I will use the 0.1 threshold as the lower limit of the trough in 3b. The results are insensitive for a 20% variation around this value, and I will discuss the implications of this choice in Sec. 4. Note that this analysis does not differentiate regions of high temporal variability based on the distance from the continent, as for instance done in (Stroeve et al., 2016) with the SIC threshold criteria. Regions of high temporal variability showing MIZ-like conditions can also be found in the interior of the sea ice, as it will be further analysed in Section 3.3. It is remarkable to note that the heavier filtering and gap-filling used in the standard NOAA/NSIDC CDR version 4 and OSI-450 introduce a smoothing in the distribution of the median that flattens the second peak and removes much of the variability in the MIZ (Fig. S1 and S2 in the supplementary material).

The NOAA/NSIDC CDR $\overline{\sigma}_{SIA}^n$ computed in eq. (3) is shown in Fig. 4 as an overall climatological indicator of SIC variability in the Southern Ocean. The standard NOAA/NSIDC version 4 and OSI-450 are substantially equivalent but with less noise associated to values lower than 0.1 in the open ocean region (see Fig. S3 in the supplementary material; the OSI-450 product also leads to slightly smaller values of the $\sigma_{SIA}$ climatology at the ice edge because of the use of a stronger open-ocean filter). The extent of the regions presenting MIZ features increases from November to December in a diffused fashion. Later in the austral winter season, these regions are confined within a band around the sea ice edge that progresses northward and shrinks at the boundary with the ocean. The higher values and the largest meridional spread are found in April and May in the Weddell and Ross Seas. In June and July, the large expanse of the Eastern Weddell Sea between 15°W and 40°E corresponding to the Atlantic bulge of the sea ice edge is characterised by large SIC variability that extends towards the continent. The value of the indicator can also be appreciated by looking at how it captures the variability corresponding to the Maud Rise polynya. The impact on SIC variability in this area is visible from September throughout November, with the latter characterised by a climatological value above 0.2 over a large expanse of the sea ice covered region. In November, this region denotes a decrease of the indicator, because the polynya is usually fully developed and the open ocean traits prevail.

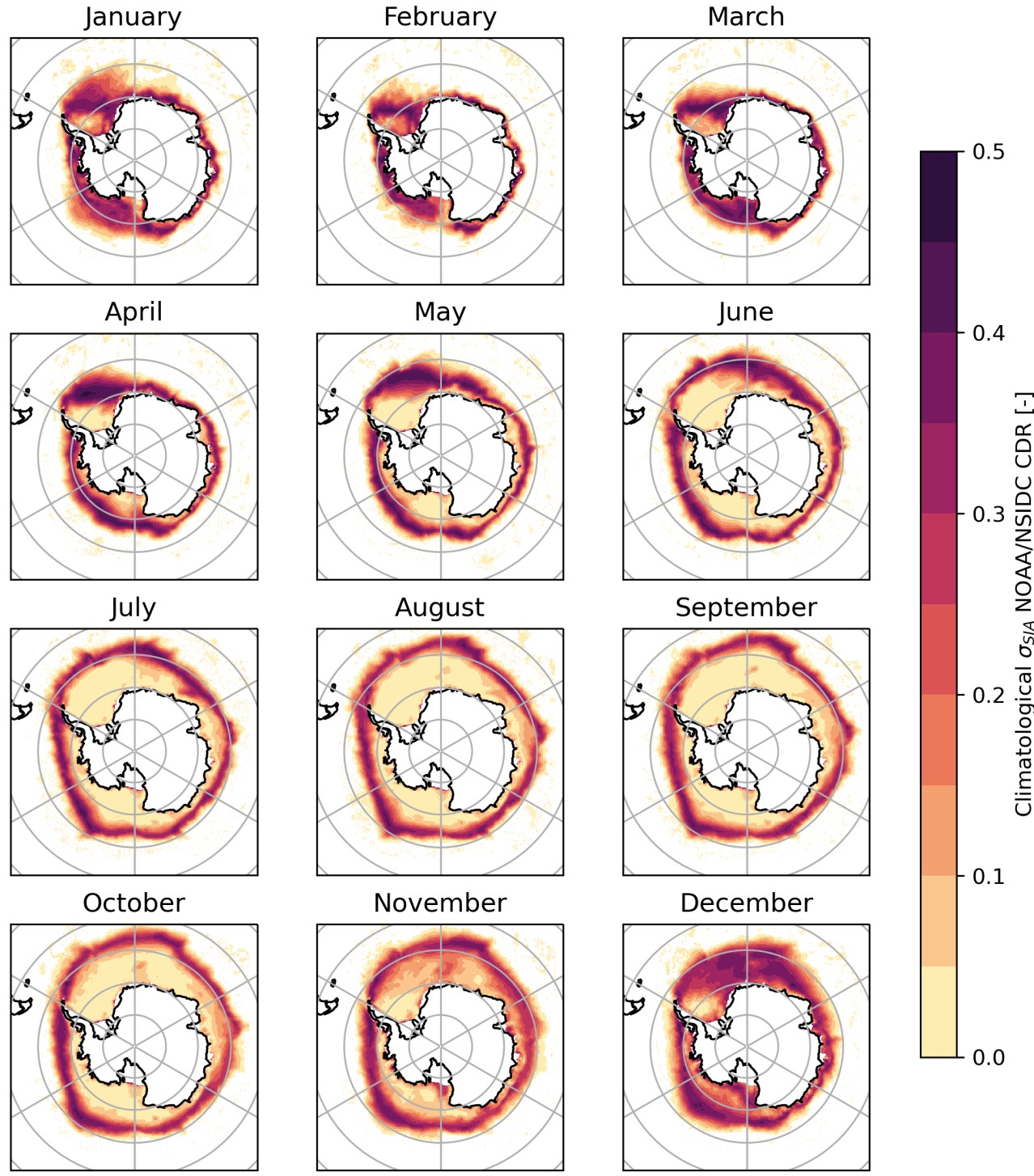

**Figure 4.** Climatological values of the indicator $(\overline{\sigma}_{SIA}^{n})$, computed as the standard deviation of the daily anomalies for each month in the whole time series (see eq. 3).

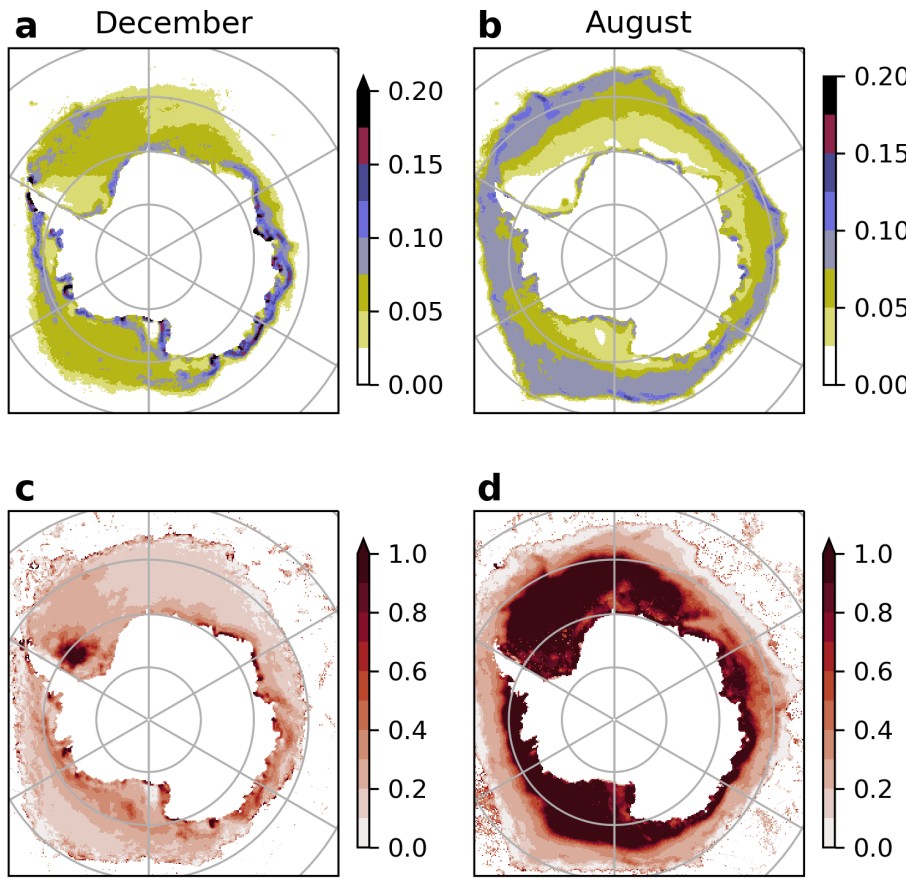

**Figure 5.** Comparison of the spatial and temporal variability for the NOAA/NSIDC CDR. a, b) climatological spatial standard deviation for the months of December and August; c, d) ratio between the spatial standard deviation shown in panels a and b and $\overline{\sigma}_{SIA}^{n}$ for the same months of December and August. All the months are shown in Figure S4 of the supplementary material.

## 3.2 Assessment and regional analysis

The climatological maps are useful to highlight the seasonal features of the MIZ, which will be further analysed in the next section. However, it is relevant to first appreciate the uncertainty associated with the assumptions of the indicator, and analyse how it differs from the more traditional analysis based on the operational SIC threshold. One of the assumptions is that MIZ conditions are more evident as temporal changes over the monthly scale at any given observable point. Antarctic sea ice is highly variable at a variety of scales, and this variability can be distinguished in terms of temporal variability at a given location and spatial variability over a certain region. An ergodic process is characterised by its time mean being equal to the ensemble (spatial) mean over a given temporal and spatial ambit. In an ergodic process, space and time variations are interchangeable. Sea ice can be modelled like an ergodic process (Hogg et al., 2020), and this assumption is also made when detecting variability

from multi-model ensembles (e.g. Horvat, 2021). The Antarctic MIZ is however largely under-sampled, and there is limited knowledge on whether time and space variability are equivalent. To check if $\sigma_{SIA}$ is an indicator of physical variability in the sea ice, I have compared it with the estimated spatial uncertainty from the NSIDC/NOAA CDR (Sec. 2.2). The mean climatological values for the months of December and August are shown in Fig. 5a-b, chosen as examples of austral summer and winter months before the months of minimum and maximum extent. In summer, the mean spatial standard deviation of the sea ice cover fraction is below 0.1 almost everywhere but in the regions of coastal polynyas. In winter, the highest spatial variability is found at the edge, corresponding to the MIZ region. Panels c and d in Fig. 5 show the ratio between the spatial variability and the $\overline{\sigma}_{SIA}^{n}$ indicator from Fig. 4. This ratio is lower than one, in the range 0.1-0.3, for the large majority of the ice-covered ocean, besides the pack ice region in August. Mean temporal variability thus exceeds spatial variability in the MIZ region in winter, also hinting at a dominance of local temporal variability in the extended summer MIZ. I also notice that the standard deviation of the anomaly used in the definition of $\sigma_{SIA}$ is a lower-range estimate of variability, since it captures the inter-annual component. The same analysis performed on the spatial standard deviation would likely lead to smaller values, further lowering the ratio in the MIZ regions. This relationship holds for all the other months, as shown in supplementary Fig. S4.

A main question is weather the proposed indicator performs 'better' than the operational definition of the MIZ. I argue that this question cannot be adequately answered for two main reasons: 1) the use of a threshold-based MIZ has not been objectively assessed in the literature but merely applied operationally, which poses a considerable challenge when proposing any alternative indicator; 2) there are no ancillary observational datasets (at least not derived from passive microwave measurements) that would allow an independent assessment of any metrics. MIZ diagnostics are usually applied in climatological or integrated analyses (for shorter times and specific regions, SIC is the variable of preference), and as such it is difficult to assess them against local ship observations or SAR images. However, these points should not dissuade us from comparing with data that have sufficient time coverage, as for instance buoy data lasting longer than a month, or comparing the different metrics without a benchmark, as typically done in model intercomparisons projects.

I offer two examples to demonstrate the advantages of this diagnostic. Womack et al. (2022) analysed the trajectory of an ice-tethered, non-floating buoy that drifted through the marginal zone in the East Antarctic sector for more than 5 months from July to the beginning of December 2017. The study indicated that the sea ice was permanently in free-drift conditions with SIC close to 100%, showing a high correlation between the sea ice drift and the wind direction, as well as various loops in the trajectory in correspondence with the passage of extratropical cyclones. The paper focused on the daily changes in SIC and the buoy distance from the edge. In this example I compare the pathway of the drifter over each month against the average monthly location of the threshold-based MIZ location and the map obtained with the $\sigma_{SIA}$ indicator. We observe that in winter there is good correspondence between the two diagnostics, as further shown in Sec. 3.3 and Fig. 10, but the shaded field indicates that SIC has been more variable in the interior of the sea ice where the buoy drifted, as well as in the outer edge in December when the buoy sank. The MIZ was not homogeneous in July and August, and although this variability did not show in the SIC values at the buoy location, the spots of high $\sigma_{SIA}$ values indicate the presence of synoptic activity at the margin (Vichi et al., 2019; Womack et al., 2022) that resemble the trajectory of the buoy. September and October were quieter, although we still

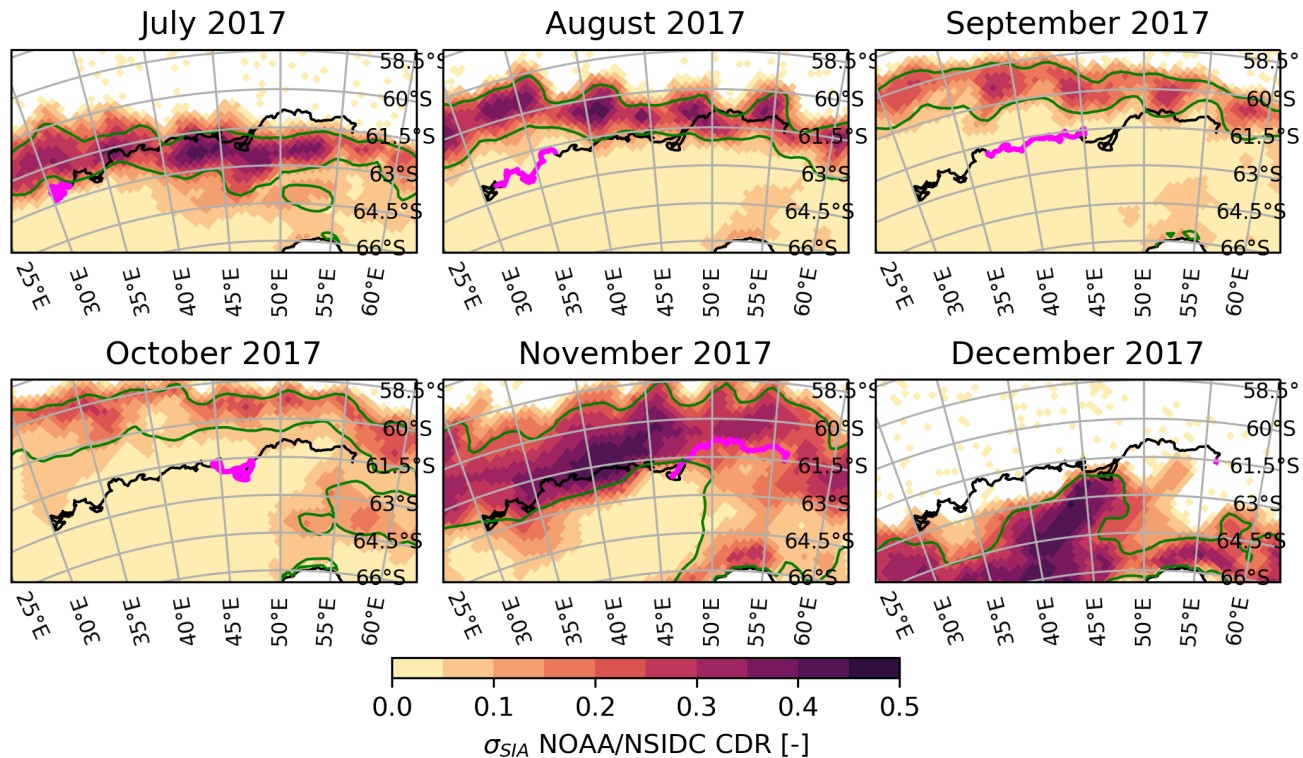

**Figure 6.** Trajectory of the ice-tethered, non-floating drifter studied by Womack et al. (2022) in winter 2017 (from 04-07-2017 to 01-12-2017, black line) overlain to the $\sigma_{SIA}$ indicator field (shading) and the 0.15-0.80 SIC range (green contours) from the NOAA/NSIDC CDR. The magenta lines indicate the paths followed during each month.

observe high intensity at the margin that coincide with the meandering of the trajectory. Such details cannot be obtained with the analysis of the MIZ contours alone, because it is difficult to trust a bending of the 0.80 contour level, while the confidence increases when it is associated to consistent areas of intense variability.

As a further example of intercomparison with the operational MIZ definition, Fig. 7 shows that the proposed indicator is sensitive to inter-annual variability in months that have been reported as anomalous, and with more details than they can be derived from the threshold-based MIZ. November 2016 was very anomalous with respect to the previous years in terms of SIE (Turner et al., 2017; Parkinson, 2019), and this has been captured in the threshold-based MIZ extent (shaded region in Fig. 7a). However, looking at the same year in panel b, the whole Atlantic sector was characterised by intense and extended MIZ-like conditions not only in the region of the Maud Rise as indicated by the SIC thresholds in Fig. 7a, a condition that persisted until 2019. The threshold-based MIZ definition only indicates the extent, and not an intensity of the MIZ conditions, although values below 0.8 are indicative of gaps in the cover that persisted for a month. According to the indicator, there was large temporal variability also at the boundary between the Amundsen-Bellingshausen and the Ross Sea sectors, which is not visible

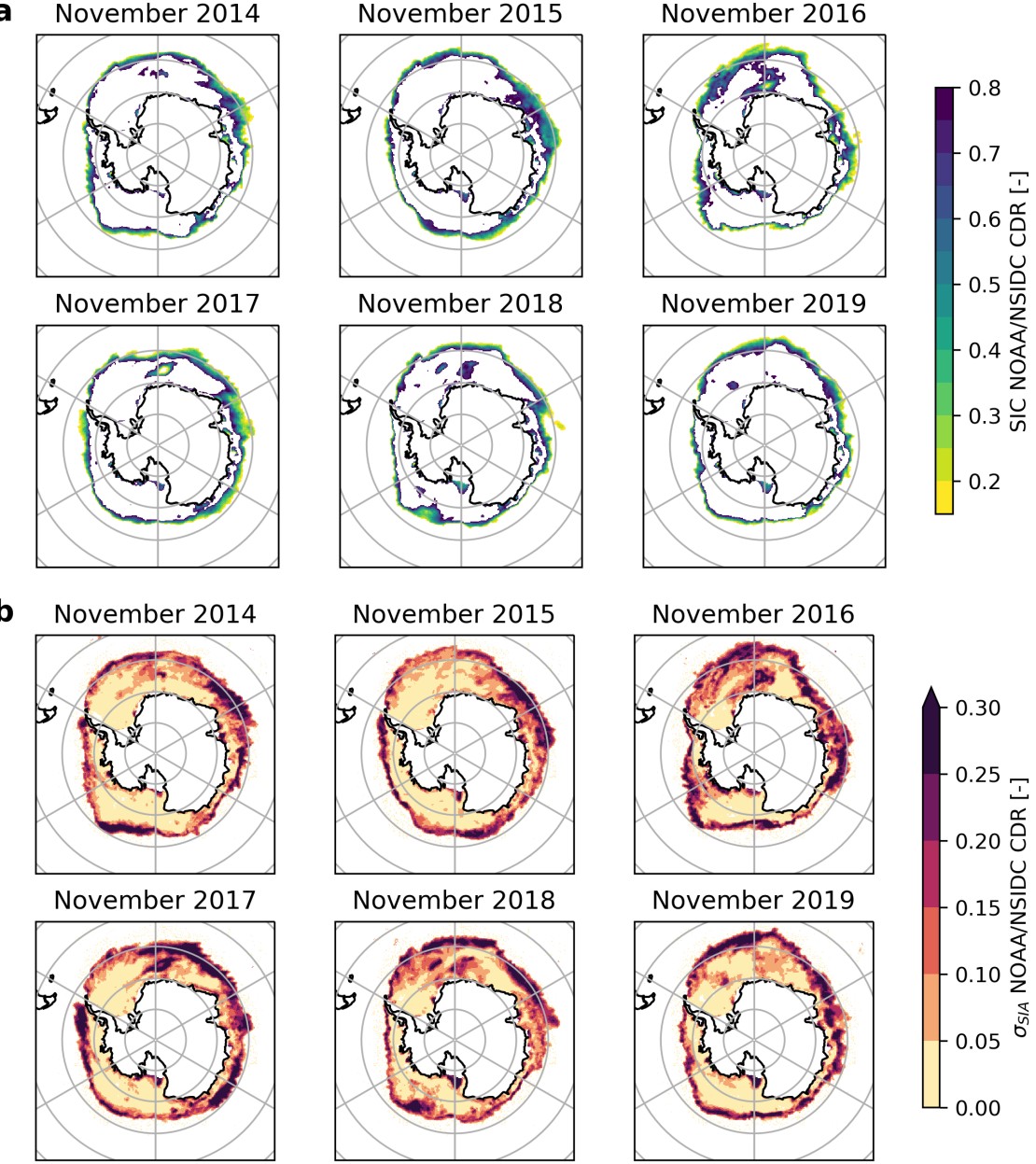

**Figure 7.** November maps of a) the mean SIC for the standard MIZ thresholds $(0.15 \leq \text{SIC} < 0.80)$, and b) the $\sigma_{SIA}$ indicator from the NOAA/NSIDC CDR for the years 2014-2019. Note the scale change with respect to Fig. 4. Years from 2008 to 2019 are shown in supplementary figures S5 and S6.

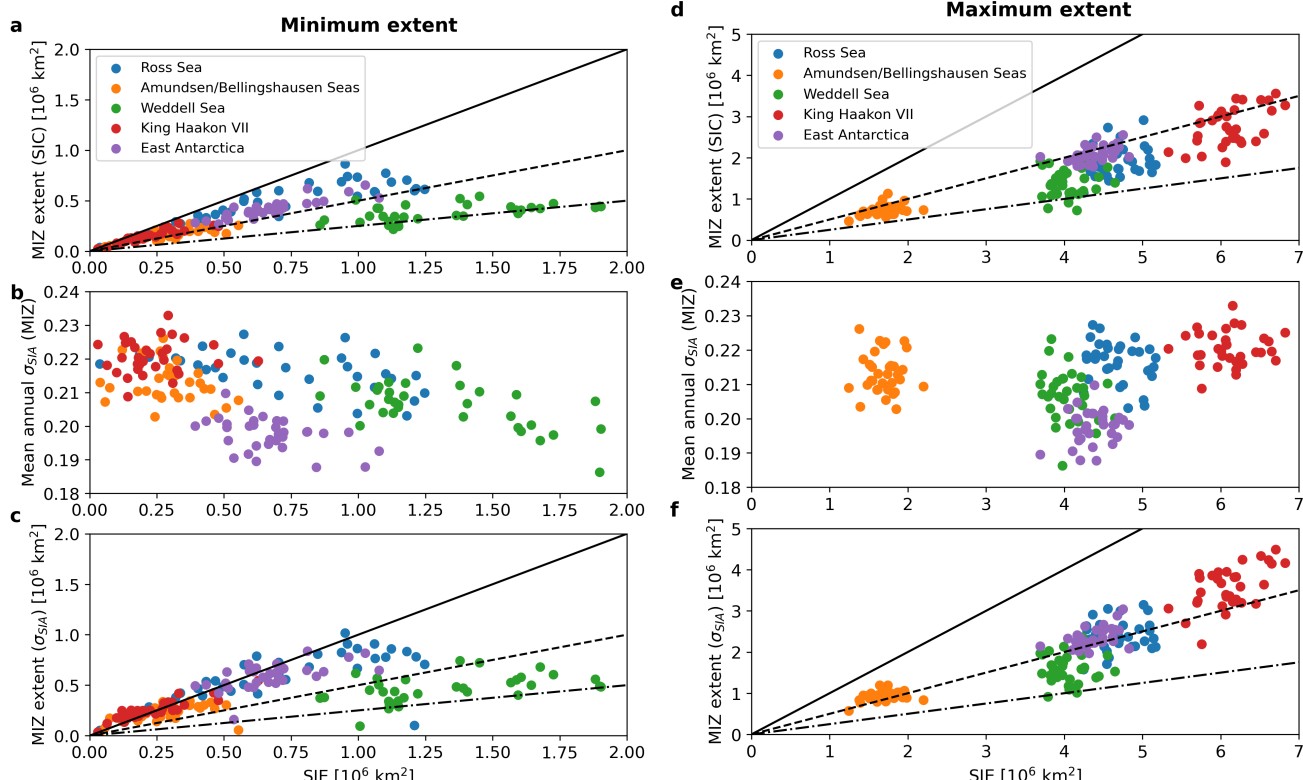

**Figure 8.** Relationship between the total minimum and maximum sea ice extent (SIE, x-axes) and three MIZ properties (y-axes) in the different sectors of the Southern Ocean for the years 1987-2019. a) minimum monthly MIZ extent computed using the 0.15-0.80 SIC mask criterion ; b) annual mean of $\sigma_{SIA}$, computed for the MIZ pixels where $\sigma_{SIA}^m > 0.1$; c) minimum monthly MIZ extent computed using the $\sigma_{SIA}^m > 0.1$ mask criterion. d-e-f) the same but for the maximum of each year. The lines represent the 100% (continuous), 50% (dashed) and 25% (dot-dash) MIZ fraction.

in panel a. In addition, Novembers 2017-18 were not much different from the earlier years before 2016 in terms of the mean SIC apart from the Maud Rise polynya in 2017, while the $\sigma_{SIA}$ analysis highlights a persistence of large temporal variability in the Atlantic sector. Similar conditions were previously observed in 2009-10 (see supplementary figures S5 and S6), which was another period of negative SIE anomalies especially recorded in the Weddell Sea and Indian Ocean regions (Parkinson, 2019, see her Fig. 3 and 4).

I have also tested if sectors with more extended sea ice are more prone to temporally variable SIC, and thus they exhibit covariance with large $\sigma_{SIA}$ values. The MIZ fraction has a complex regional relationship with the SIE, with a seasonal cycle that differs for different Antarctic regions (Stroeve et al., 2016). The sectors have been defined following Raphael and Hobbs (2014), since they proposed a separation based on large scale atmospheric drivers rather than using arbitrary longitudinal boundaries. The total maximum monthly SIE for each sector and each year in the period 1987-2019 have been plotted against

the MIZ SIE computed with the 0.15-0.80 threshold, and analysed in combination with the value of $\sigma_{SIA}^m > 0.1$, averaged over the sector and the whole year (Fig. 8a-b). This latter diagnostic gives an indication of the mean variability of the MIZ in each sector, and it can be done this way because the minimum and maximum extents fall within the same calendar year in the Antarctic sea ice season.

The various sectors show quite distinct clusters, with only some overlap. If we consider all sectors as a single cloud of points, both the minima and maxima of the MIZ extent follows a linear relationship with the total SIE (Fig. 8a,d). The Weddell Sea shows the largest SIE in summer with the largest interannual variability, and the points cluster around the 25% line. The MIZ fraction is higher in the King Haakon VII (KH), around 50%, and in the Amundsen-Bellingshausen (AB) sectors, while the Ross Sea (RS) and East Antarctica (EA) have an intermediate minimum SIE, with an MIZ fraction larger than 50%. We observe a decreasing trend of MIZ variability with the increasing SIE in summer: the sectors with low SIE like KH and AB also have a highly variable SIC, indicated by the higher values of $\sigma_{SIA}$. The EA sector does not follow the linear trend because the variability is lower than in the other regions. The SIE here is comparable to the WS during summer, but SIC departs less from the mean monthly climatology. Since the seasonal cycles are different in each sector, the clustering of the maxima shown in Fig. 8d are not the same as in panel a, and also the spread of different years is lower. During the maximum winter SIE, we note that regions with different magnitudes of SIE and different MIZ fractions have the same amount of variability. The AB, RS and KH sectors have similar ranges of the mean $\sigma_{SIA}$, although the Weddell Sea records the highest values. In general, the magnitude of SIC variability appears to be independent of the characteristics of the sectors. Only the East Antarctic sector still stands out as the region in which the MIZ fraction is extended in winter but with relatively lower intrinsic SIC variability than in the other sectors. The estimate of the MIZ extent based on counting the area of pixels with $\sigma_{SIA}^m > 0.1$ has also been computed and shown in Fig. 8c. This will be discussed in the next section together with the climatological seasonal cycle of the whole Antarctic MIZ.

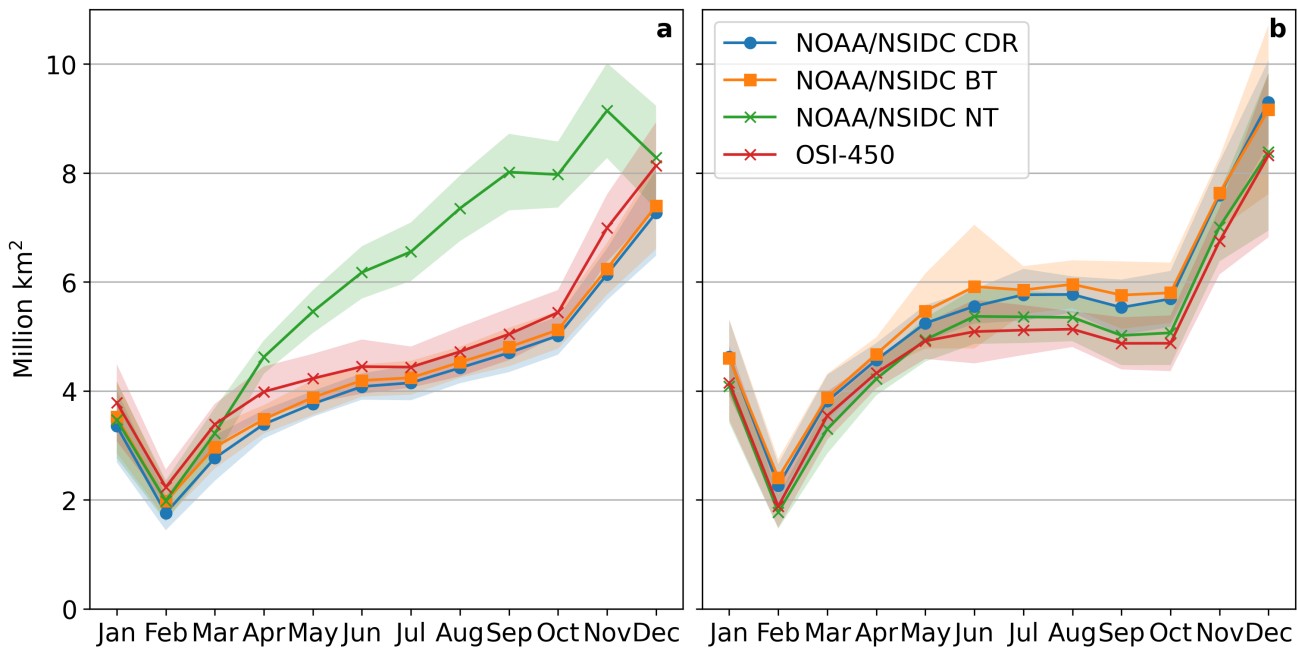

**Figure 9.** Seasonal cycle of the MIZ extent estimated from the a) SIC criterion ($0.15 \leq$ SIC $< 0.80$) and b) the $\sigma_{SIA}$ indicator ($\sigma_{SIA}^m > 0.1$). The results are shown for all the products described in Sec. 2.1.

### 3.3 Patterns of seasonal variability

The above analysis highlights that, due to the intrinsic seasonal nature of Antarctic sea ice, there are wide regions where SIC shows high inter-annual variability in any month of the year, which is only partly captured by analysing the mean monthly concentration. To strengthen this concept, the seasonal cycle of the circumpolar MIZ extent was computed from the monthly indicator $\sigma_{SIA}^m$ for every year and then averaged. This measure is comparable to using the mean monthly SIC comprised between 0.15 and 0.80 to compute the extent. Following from the results shown in Sec. 3.1, any pixel with a value of the

indicator larger than 0.1 was assumed to be characterised by MIZ processes and included in the spatial integral. Fig. 9 shows the comparison between the MIZ extent computed using the 0.15-0.80 SIC criterion and the one proposed here, for all the products described in Sec. 2.1. As previously shown by (Stroeve et al., 2016), the MIZ area based on the SIC threshold is largely affected by the retrieval algorithm. In their analysis, the NT algorithm had a higher MIZ area than BT and a larger proportion of inner open-water ice and coastal polynyas, which are included in this MIZ threshold-based criterion, and hence

the absolute value shown here is higher than the one presented in Stroeve et al. (2016, their Fig. 5). The $\sigma_{SIA}$-based MIZ extent is instead independent of the algorithm choice or product, because the relative variability is equally captured. We notice that in the NOAA/NSIDC V4 product the BT and CDR estimates are very similar (Fig. 9a), while the threshold-based estimate of the MIZ extent computed with the CDR from V3 was much lower (Fig. S7). The threshold-based MIZ seasonality (Fig. 9a) grows linearly from summer to spring, where it increases sharply until the peak in December. The inter-annual spread indicated by

the shaded area is similar throughout the year, apart for the NT product that increases in winter and spring. The cycle obtained from the $\sigma_{SIA}$ indicator shows a greater increase from February to May, and then the MIZ extent remains constant, but more variable from year to year. This alignment of the BT and NT products is not a result of the climatological averaging, as shown in Fig. S8. We also note that in the anomalous 2016, the progression of the MIZ extent was more linear and similar to the threshold-based climatology. The November MIZ extent was still in the range of the previous years, while it collapsed in

December.

    This indicator quantifies the intensity of temporally variable MIZ conditions, as opposed to the SIC range criterion, which returns a binary mask based on the average concentration. A climatological MIZ mask can still be obtained by considering the pixels that are climatologically more likely to present MIZ features (with values of $\overline{\sigma}^n_{SIA} > 0.1$, Fig. 10). Here we observe that the two criteria are more similar in the winter to early spring months from July to October. However, the $\overline{\sigma}^n_{SIA}$ MIZ mask

is generally wider and more extended both onto the open ocean and into the pack ice. This is more evident in the Eastern Weddell Sea from 0-50°E, and also in the Ross Sea between 120-160°W. This difference increases from October to June, with a peak in November. The latter is indeed the month that has shown the largest variability in the records (Turner et al., 2017), as previously highlighted in Fig. 7. This increase of the MIZ extent is also visible in the regional analysis of the minimum and maximum MIZ extent obtained with the same method and shown in Fig. 8c,f. In the month of minimum extent (February), all

sectors show a higher fraction of pixels with MIZ features, with the MIZ fraction also exceeding 100%. There is also more year-to-year variability in the Weddell Sea MIZ extent than with the SIC range criterion (Fig. 8a). However, the relationship between the sectors is unchanged. It should not be surprising that the MIZ extent presented in this work exceeds the total SIE. This is because this method detects pixels that are statistically more likely to be affected by changes in SIC from year to year, rather than the pixels that had an average monthly mean of SIC>0.15. Antarctic sea ice can drift quickly in a short period of

time, and for a few days over a month. This would temporally change the concentration but it will less likely affect the mean variability unless these changes occur several times. This indicator has been specifically designed to capture this property, and to give a likelihood of encountering MIZ conditions, as it will be presented in the next section.

    There is finally a fundamental computational difference between the climatological averaging of the monthly extents shown in Fig. 9b, in which a monthly mask is multiplied by the pixel area then integrated over space and averaged over the years,

and the mask based on the climatological monthly standard deviation of the daily anomalies $\overline{\sigma}^n_{SIA}$. This is because the average of the standard deviations computed from sub-samples of a population is different from the standard deviation of the whole population. Note that this difference also applies to the computation of the extent using the SIC range criterion. Hence, the climatological MIZ extent shown in Fig. 9 is an underestimation of the sea ice area that may statistically present MIZ characteristics. This is graphically shown in Fig. 11, where the climatological sea ice extent (SIE) is computed from the SIC criterion

and $\overline{\sigma}^n_{SIA}$ indicator (the area of the yellow-shaded region in Fig. 10) using the NOAA/NSIDC CDR. The MIZ SIE obtained with the climatological SIC criterion (the line with the crosses) is also higher than the one shown in Fig. 9a (compare with the blue line obtained from the same product).

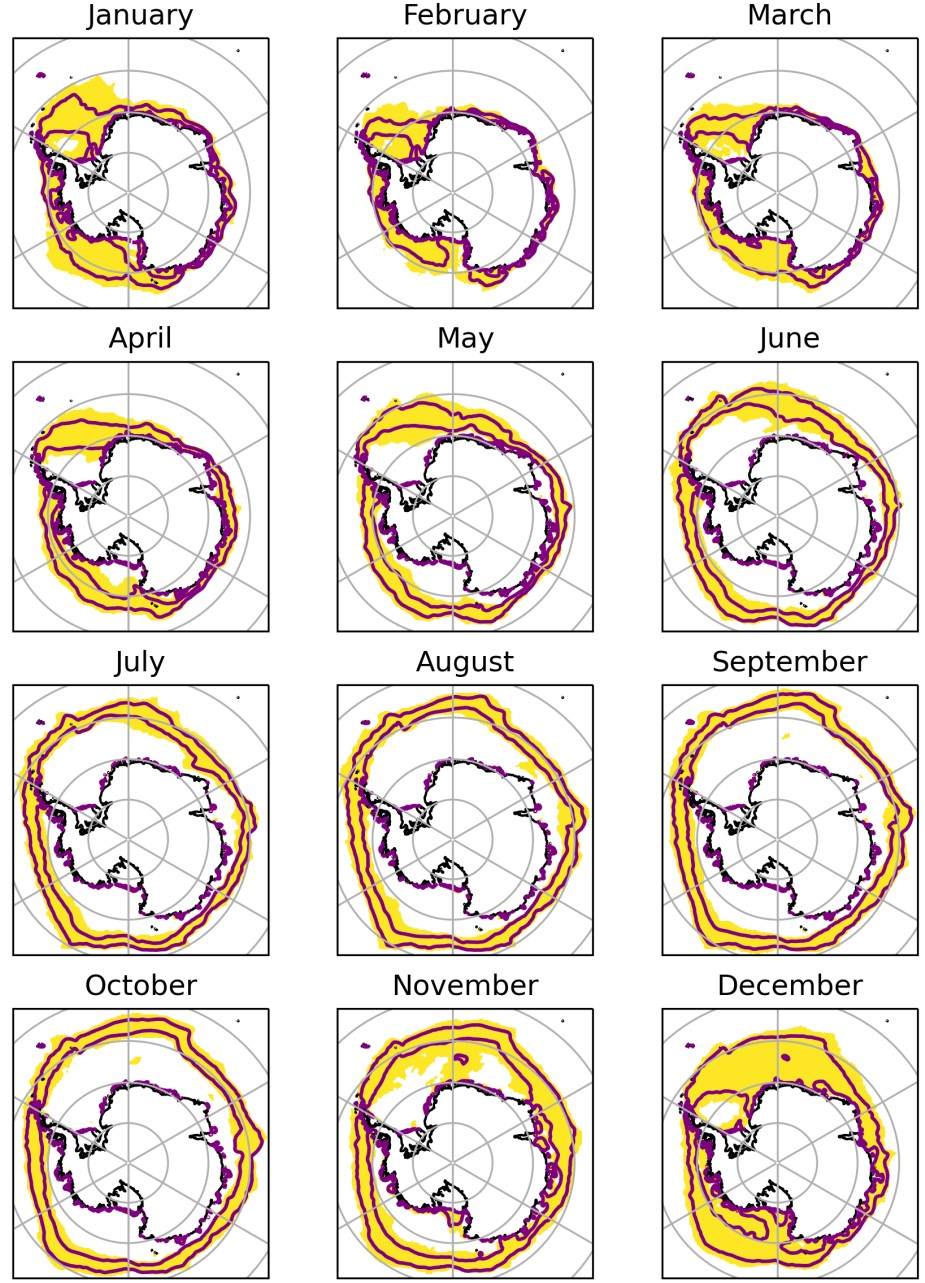

**Figure 10.** Climatological monthly mask of the MIZ obtained from the $\overline{\sigma}_{SIA}^n$ indicator. The purple line indicates the MIZ extent computed using the SIC criterion.

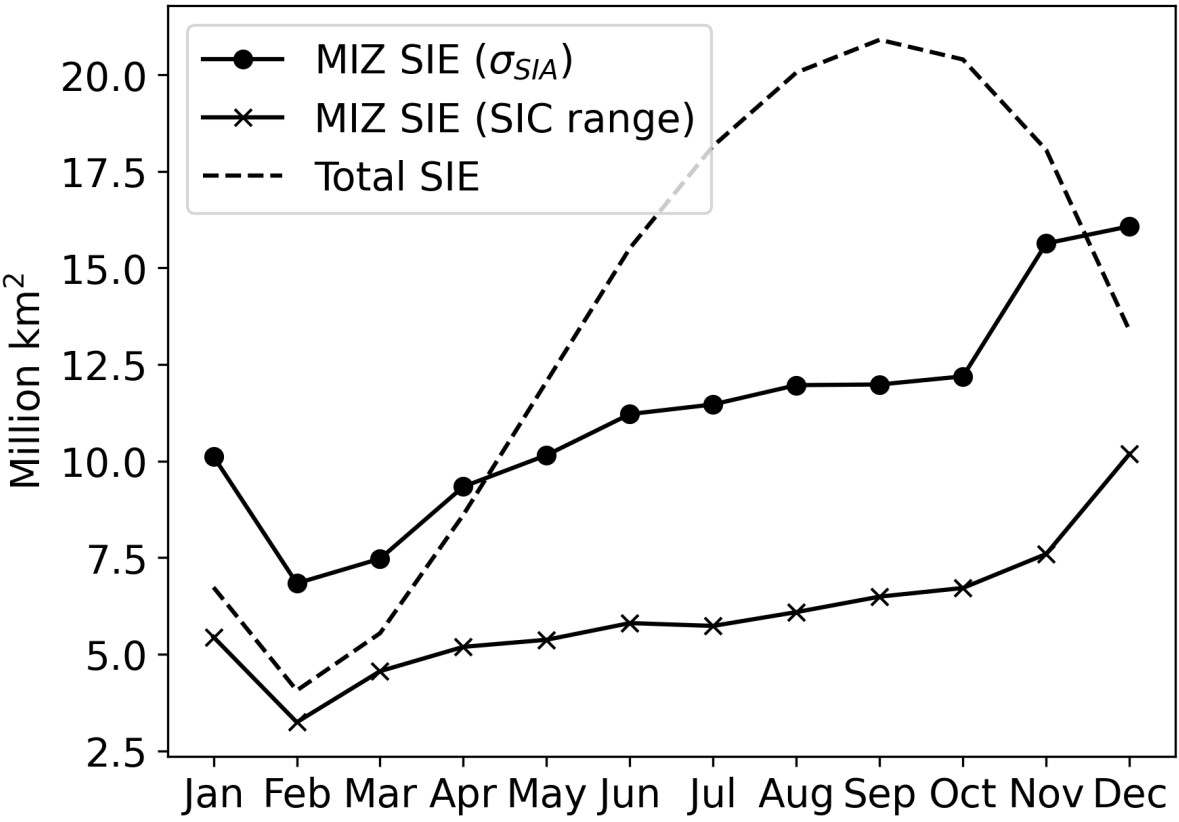

**Figure 11.** Estimated climatological extent of the MIZ and total sea ice extent computed from the monthly climatologies of NOAA/NSIDC CRD. The filled-circle line is obtained from the $\overline{\sigma}_{SIA}^{n}$ indicator using the threshold 0.1, which also includes the coastal regions.

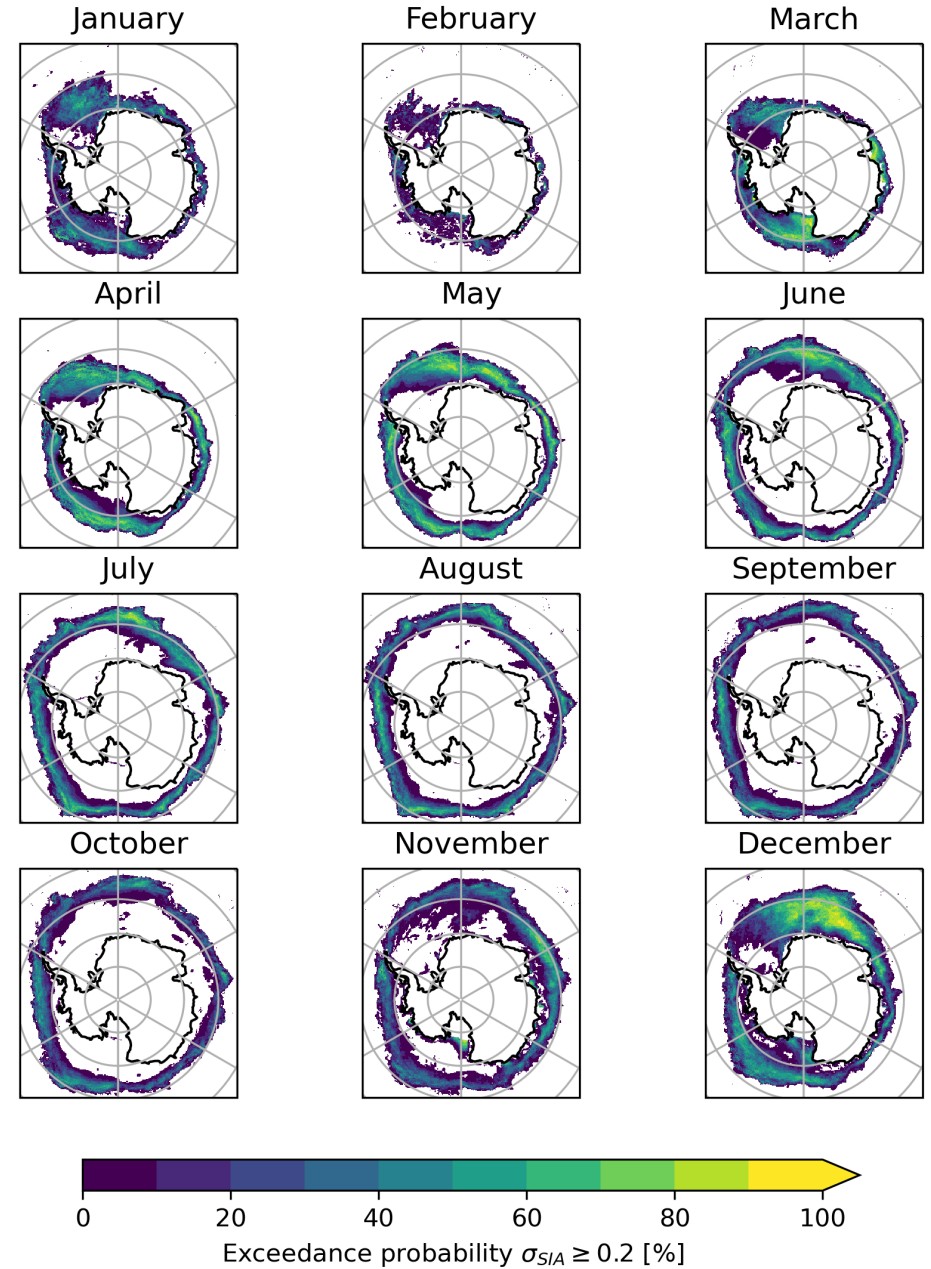

**Figure 12.** Monthly values of the exceedance probability for a threshold $\sigma_{SIA} = 0.2$ from the NOAA/NSIDC CDR.

## 3.4 Exceedance probability of encountering MIZ conditions

The previous analysis revealed that MIZ-like features in Antarctic sea ice are not necessarily confined to the outer edge or to coastal polynyas, but they can also extend to the interior of the pack ice. It is therefore of interest to quantify the likelihood of encountering MIZ conditions in a selected month. The probability of exceeding a given value of variability according to the method presented in Sec. 2.2 is shown for the substantial threshold $\sigma_{SIA} > 0.2$ in Fig. 12 (the maps for the $\sigma_{SIA} > 0.1$ are shown in Fig. S9 in the supplementary material). The presented value is twice the threshold used in the previous sections, to assess the probability of encountering extremely variable sea ice states ($\sigma_{SIA} = 0.2$ is the 99th percentile of the distribution of medians shown in Fig. 3).

The exceedance probability is different from month to month in different sectors of the Southern Ocean (Fig. 12). This is consistent with the lack of consistency when comparing regional and hemispheric values of SIE trends. For instance, the Ross Sea presents the highest chance of finding variable sea ice state in March over the entire region, while in December this is more likely in the Weddell Sea and in the Indian Ocean sector up to 90°E. The regions where the extent of sea ice from the continent is narrower, such as Eastern Antarctica, tend to show less variability in the sea ice state. In more accurate terms, the probability of exceeding a high value of the indicator in East Antarctica sea ice is lower with respect to the other regions, but the whole sea ice covered region should be classified as MIZ, since the probability of exceeding the 0.1 threshold is above 80% in every month (supplementary Fig. S9). This region is therefore one of the most interesting to capture the seasonal processes at the air-sea ice-ocean interface, because the MIZ remains confined within the same latitudinal band throughout the year. Combining this information with the analysis presented in Fig. 8, we may conclude that there is lower year-to-year intensity of variability in East Antarctica (in terms of the magnitude of the anomalies), but that the sea ice state is in a permanent MIZ condition.

In general, there are lower chances of exceeding the threshold value both in the outer edge and in the internal pack ice. This feature is caused by two different processes. At higher latitudes (mostly in autumn and winter), it is less likely to find variable conditions because the sea ice advances so far north only in a few years. February is an interesting month, because almost in every pixel in which sea ice has been observed in the satellite records there is a similarly low probability to exceed the threshold. This means that there are small chances of encountering brash ice but it is more likely that open drift conditions will be prevailing. At lower latitudes, on the other hand, the probability close to zero is because there are persistent pack ice or polynya conditions (the white regions between the coloured sectors and the Antarctic continent). They can be found in all months but March, which only shows the few regions of multi-year ice in the eastern Weddell Sea and the Ross Sea polynya. These are regions where consolidated conditions and sea ice features that are more likely to be similar to the Arctic are found according to the satellite records.

June and July are instead the months of higher chances of encountering SIC variability away from the edge towards the interior of the Eastern Weddell Sea, and hence a sea ice state that is more typical of the MIZ. In these regions, assuming pack ice conditions in numerical models and other conceptual considerations may lead to an underestimation of the air-sea exchanges. The SIC values may be generally close to 100%, but the fluctuations around this value are large, which is indicative of a sea ice state that is affected by boundary processes.

## 4   Discussion and conclusions

### 4.1   Towards a multivariate definition of the MIZ

This work aims at reviewing the way we consider the Antarctic MIZ, shifting the perspective from considerations based on the absolute concentration to the relative temporal variability. This is seen as one way to overcome the difficulties of detecting a clear relationship between concentration and ice type in the Southern Ocean. The MIZ should be defined in terms of the physical processes that shape the type of ice and its stages of formation and decay, from pancakes, to grey-white ice into young and first-year ice. Unfortunately these properties cannot be derived at relatively high frequency and large scales, therefore SIC from space should be further exploited to give insights on the variability of Antarctic sea ice instead of just its mean state. The use of absolute SIC thresholds does not tell the whole story of the MIZ seasonal cycle, and especially it does not give a direct measure of the temporal variability, which I demonstrated to be a characteristic that dominates over the spatial variability in the MIZ (Fig. 5).

The proposed method is complementary and extends the traditional threshold-based definition, hinting at the importance to use a multivariate approach for the MIZ definition that combines mean and variance. It allows to identify regions of higher variability and to quantify the climatological relative intensity. It gives a quantitative measure of the sub-seasonal variation in SIC, and not only a binary map as it can be obtained with the threshold-based MIZ definition. The method is derived from the standard deviation of the daily anomaly with respect to a monthly climatology, a common diagnostics in the climate sciences. This indicator can be translated into maps of exceedance probability, hence giving a quantitative description of the likelihood of finding MIZ characteristics. The method does not require a priori ranges because the separation between pixels of low and high variability is obtained through a distributional analysis that reveals a bimodal pattern in the Antarctic.

A threshold is nevertheless necessary, which is defined as the trough that distinguishes points of low variability, which are more typical of the inner pack regions, from the more variable MIZ regions. This implies that conditions of high variability similar to the ones found at the margin can also be found in more consolidated sea ice regions from a climatological viewpoint, in agreement with the observed penetration of waves deep into the pack ice (Kohout et al., 2014, 2015; Stopa et al., 2018; Massom et al., 2018; Vichi et al., 2019; Kohout et al., 2020). Whether this variability has to be attributed to the incidence of extratropical cyclones that stimulates daily SIC changes is currently being investigated. Intense cyclones have a systematic statistical association with atmospheric temperature extremes over Antarctic sea ice (Hepworth et al., 2022). However, the same study reports that moisture extremes are more associated with atmospheric rivers at the sea ice margin, and there is still a large portion of extreme atmospheric events in the interior of the pack ice that cannot be related to the presence of cyclones. Whether extreme atmospheric events are needed to engender variability in the pack ice that persist at the climatological scale is still an open question, and this same indicator can be applied in this context by considering anomalies over the synoptic time scales.

One main concept of the methodology presented in this work is the use of daily SIC anomalies derived from the climatological monthly mean. This is based on the evidence that Antarctic sea ice has a clear seasonal pattern (Eayrs et al., 2019) but high variability from year to year and uncertain trends in different regions (Matear et al., 2015; Yuan et al., 2017; Parkinson, 2019). I

remark that this indicator is explicitly constructed to combine the sub-seasonal variability due to the advancement and retreat of sea ice, as well as the smaller scale changes in response to the synoptic weather, such as the passage of extratropical cyclones. Regions with higher mean variability like the King Haakon VII sector (Fig. 8) are indeed those with the higher incidence of extratropical cyclones, and where sea ice trends are likely driven by the weather (Matear et al., 2015; Vichi et al., 2019). The same indicator has been applied to the Arctic, where the ice cover fraction is used as an indicator of the ice type. The analysis of the distribution median shows a much larger density of pixels with low temporal variability (multi-year and thicker pack ice) and a less extended tail with higher SIC variability. This confirms that SIC has smaller sub-seasonal variability than in the Antarctic, and the use of the threshold-based definition is likely sufficient to capture the regions where MIZ processes occur. Nonetheless, this indicator could be useful in studying transitional seasons and could be applied to different periods to assess whether there has been an increase in sub-seasonal variability with the increased Arctic sea ice loss.

The results presented in Fig. 9 have shown that this indicator removes the mismatch in the estimation of Antarctic MIZ extent using different algorithms (Stroeve et al., 2016). The CDR and the BT NOAA/NSIDC threshold-based estimates cluster together with the OSI-450 product, while NT is much larger. This was not the case with the NOAA/NSIDC V3 (Fig. S7a). This mismatch is not visible with the $\sigma_{SIA}$ estimates, which indicates that the threshold-based definition is sensitive to the specific data processing, while the variability captured by each data product remains the same.

## 4.2 Caveats and future applications

A note of caution is necessary to clarify the difference between the MIZ extent derived with the 0.15-0.80 range and the climatological mask obtained through the $\sigma_{SIA}$ indicator (Fig. 10), as well as the related seasonal cycle (Fig. 11). This masking method (where pixels with $\sigma_{SIA} > 0.1$ are classified as MIZ points) is complementary to the MIZ SIE and should not be used for computing the marginal ice zone fraction (like in Horvat, 2021). The SIE criterion must be the same for both the MIZ and the total ice cover. However, the study of the MIZ fraction may be less sensitive in the Antarctic, and a comparison between model outputs and satellite data using this indicator may give interesting insights. There is no incongruence in the mismatch between these estimates of the MIZ extent, because they measure different properties. In terms of the seasonal climatology, the MIZ area obtained through the use of a fixed threshold slowly grow in winter, and is below the total summer area (Fig. 11). The estimated MIZ area using the indicator reaches a plateau during the austral winter months and is instead more extended in summer than the total SIE. This may seem paradoxical, because the region classified as MIZ cannot be larger than the sea ice extent. This is however an artefact of the use of climatological means and the 15% baseline, which skews the distribution towards the low values, disregarding the natural large variability of Antarctic sea ice and the diversity of ice types. There are more pixels where daily SIC anomalies have values larger than 0.1 at the sub-seasonal scale in summer. This also includes the polynya regions, which, due to their nature, are more affected by daily changes in weather conditions. The proposed analysis is therefore more oriented towards the estimation of variability due to heterogeneous ice conditions, independently of where they are located.

There is no specific reason that the proposed indicator is the best method to quantify the variability. Alternative indicators could be used, such as for instance the monthly averaging of daily maps of the SIC-threshold MIZ (i.e. identified points with

$0.15 \leq SIC < 0.80$ every day) instead of defining the threshold on the monthly climatology, and hence a monthly mask. This method would indeed add a measure of intensity, but would still not detect changes when sea ice is above 0.80, a condition often found in the MIZ. This does not mean that the threshold-based estimates are not accurate, but that there are regions of the ice-covered ocean that present physical characteristics similar to the MIZ even when the SIC fraction is above 0.80. Perhaps, as shown by the buoy example presented in Fig. 6, it is in the regions around the 0.80 SIC level that most of the missed variability

is found. However, the drift data indicates high mobility in areas where $\sigma_{SIA}$ is still not high enough, which indicates that this variability is probably more active at the synoptic time scales.

  To conclude this discussion, I would like to offer a critical analysis on the use of SIC products that have been optimised for climate studies, like the CDR presented in this work. SIE, as an essential climate diagnostics, has been designed to be a smooth measure of long-term climatic variability. My perspective is focused on Antarctic applications, which are less likely to

be supported by direct observations, and methodologies developed for the Arctic tend to be used in Antarctica with a necessary limited validation. The same method proposed here is indeed supported by a few examples only, because a more systematic analysis is not yet possible. Users may not always be aware of the subtleties of the satellite-derived products. For instance, the differences between NOAA/NSIDC CDR V3 and V4 are substantial, as shown in the supplementary figures. The choices of filling gaps and enhancing the similarity with other products (Windnagel et al., 2021) are very legitimate, but users may

imply that these products and versions are interchangeable. A variety of derived products should then be made available to the users, as proposed by Lavergne et al. (2022), to allow for different types of analyses. On the other hand, it is known that SIC from space is prone to major assumptions and corrections, because the algorithm often exceeds the unit fraction and it is truncated to 1 (Kern et al., 2019). This has implications for the analysis of variability carried out here. Some variability may be dampened by the filters, or even enhanced, especially when spurious values close to 0 are not eliminated. SIC from space is

the result of an empirical model applied to selected bands of the passive microwave spectrum with few tie points, and as such it cannot encompass all the ice types found in the polar ocean. In addition, day to day variability is biased by the construction of composite of different swaths to obtain a daily picture of the sea ice distribution. As suggested by Kern et al. (2019), the use of non-truncated datasets would enhance the analysis of variability. However, estimates of threshold-based MIZ extent using these datasets are not yet available in the literature, and this application goes beyond the scope of this work.

The results presented in the previous sections have several applications. They introduce a broader perspective for assessing the predictability of ice conditions for forecasting, operational activities and also as a diagnostic to evaluate climate model capabilities to simulate the adequate conditions for ecosystem studies (e.g. Williams et al., 2014; Rogers et al., 2020; **?**). Due to its construction, the method should be mostly applied in climatological or medium to long-term investigations of ice variability. Here the full record has been used to define the climatology, but climatological baselines for different periods can be

computed to detect changes in the variability with time. It may also be used in an operational context, for instance comparing each daily anomaly against the long term distribution of anomalies in a particular region. This has not however been verified in this analysis and would deserve dedicated work. From an operational view point, the exceedance maps can be used to plan scientific and logistical activities in seasonally ice-covered Southern Ocean waters. This method should not be used to measure the extent of pack ice conditions, because multi-year ice is not counted due to the high-persistence and reduced inter-annual

variability. Finally, the possibility to see patterns of intensity within the region classified as MIZ, would allow to identify further linkages with the atmospheric boundary layer, as for instance looking for associations between regions of high synoptic variability and corresponding changes in the character of Antarctic sea ice.

*Code and data availability.* The EUMETSAT OSI-450 CDR product is available at https://doi.org/10.15770/EUM_SAF_OSI_0008 and the NOAA/NSIDC Climate Data Record of Passive Microwave Sea Ice Concentration, Version 4 can be downloaded from https://doi.org/10.7265/efmz-

2t65. The code used to process the data and produce the figures is available at https://github.com/mvichi/antarcticMIZ.git. DOIs for the code and the post-processed data used in the analysis will be minted through the ZivaHub repository at the University of Cape Town if the manuscript is accepted.

*Author contributions.* Sole author

*Competing interests.* The author declares no competing interests.

*Acknowledgements.* This work has emerged from a collaboration of multiple projects. It has been supported by the National Research Foundation of South Africa (South African National Antarctic Programme and Earth System Science Research Program), the whalesand-climate.org project and the CRiceS project, funded by the European Union's Horizon 2020 research and innovation programme under grant agreement No 101003826. I would like to thank the NOAA/NSIDC support team for the prompt assistance and help in re-processing the CDR V4 dataset. I am also grateful to the anonymous reviewers who gave critical insights on the earlier version of this manuscript.

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
