# Peer review of "An indicator of sea ice variability for the Antarctic marginal ice zone"

_The Cryosphere, 2021_

## Author Response (AR1)

**Letter to the Editor and the two anonymous reviewers**

I would like to thank the Reviewers and the Editor for their patience with this prolonged revision process. The work has been more challenging than expected due to the version change of the satellite data used in the first submitted manuscript. The newly released version 4 of NOAA/NSIDC Climate Data Record of Passive Microwave Sea Ice Concentration ( https://doi.org/10.7265/efmz-2t65) is substantially different from version 3. This version is also published with some bugs in the structure of the data, which will likely be resolved soon by the NSIDC team. They have been very responsive and helpful, and I have created a set of scripts that will create a processed version that removes all the default temporal and spatial filters introduced in version 4. Users would not be able to reproduce the same results from the pre-print, and the version 3 dataset is not available anymore. Despite these corrections, the results from version 4 are not identical to version 3, although they confirm the main findings. The results from version 3 are presented in the supplementary material, when needed for the discussion. I have then added a section that critically analyse how changes in climate data records may affect other types of analysis. The new section 4.2 (Caveats and future applications) discusses the implications of having products that both satisfy the climate community interested in larger scale integrated diagnostics like sea ice extent, as well as the community looking into the processes underlying the Antarctic sea ice variability.

As a consequence of these changes, and to address the comments of the reviewers in an exhaustive manner, the manuscript is now longer than the first submission. Through the various comments, I have realised that the literature on Antarctic MIZ does not offer a complete background of the need for a complementary diagnostic of the MIZ, which would ultimately lead to a more accurate definition. I have thus changed the title to indicate that the proposed approach is meant to quantify variability of Antarctic sea ice, with a focus on the MIZ, and extended the introduction by adding three subsections that examine the current definitions, analyse the needs, and introduce the proposed methodology.

The answers to the reviewers found below are an extended version of the ones published in the discussion section of the journal. I have updated the parts for which I added further work, as well as added the references to the page numbers and sections in the revised manuscript.

Kind regards

Marcello Vichi

**RC1: 'Comment on tc-2021-307', Anonymous Referee #1, 25 Jan 2022**

This is my review of Vichi (2021).

Primarily, I want to apologize for the very long time in returning a review. I hope the author accepts my apology for this, while it is a challenging year to meet professional obligations, this simply was too much time to wait for what was a relatively short and easy-to-read paper.

In this paper, the author seeks an alternative definition of the marginal ice zone (MIZ). They use the distribution of inter and intra-monthly standard deviation of passive microwave SIC values, defining a MIZ metric as those periods where the standard deviation of SIC retrievals within a given month exceed 0.1 (unitless). Their key result is that when applied to the existing PM-SIC data, four main satellite products are in agreement as to a climatological seasonal cycle of overall MIZ extent. This is a new approach. It is clear that the MIZ requires an objective definition, and the author is making an effort to provide one.

Despite these intentions, I find methodological and conceptual flaws in the study that I do not believe permit its publication at this stage, and I recommend significant revisions be undertaken before reconsidering this MS. Generally, what this article is lacking is supporting evidence. Many of the claims made by the author about this definition *could indeed be true*, and it may have immense promise as a definition of the MIZ. But there is no supporting evidence that this definition records something physically relevant to modelers, stakeholders, or observers. With this supporting information, the paper is quite a useful and interesting contribution. But absent it, it is hard to make much of this work.

Here I give a discussion on the merits of this work, focusing on this problem of physical and statistical foundation. I am not including specific small comments because I think any revision of this MS will require substantial changes that may render such comments obsolete. Below I include two overacting suggestions which I believe should be undertaken before this paper is published.

*Answer: I would like to thank Reviewer 1 for the in-depth comments and critical appraisal of this work. This manuscript stems from the intent to ignite a discussion on the definition of the marginal ice zone with a direct application to Antarctic sea ice. The nature of the comments made me realise that, for the sake of avoiding a lengthy review of the literature in the introductory section, I failed to address the physical basis and supporting evidence of why a complementary definition of the MIZ is required in the Southern Hemisphere. I have also realised the importance of a proper definition of what I mean by ice type and variability, and how they are related to concentration, which is the only long-term information that we have available. In this revision I made a substantial effort to explain why a threshold-based indicator of MIZ characteristics does not characterise the highly variable Antarctic MIZ. The Introduction has been restructured and extended into three sub-sections:*

*1.1 Definitions of the MIZ: sea ice concentration, wave penetration and ice type*

*1.2 Characterizing variability in Antarctic sea ice*

*1.3 The need for a novel indicator*

*I also recognised the limited details given on the treatment of the statistical foundation and the analysis of errors. I have included a new section (3.2 Assessment and regional analysis), which compares the temporal variability with the spatial variability, used in the CDR as a measure of*

*uncertainty. This section offers a critical assessment of the indicator against the standard threshold-based method and presents a regional analysis as requested in a further comment.*

**RC1:** The study's motivation is that existing ways to define the MIZ do not capture the physical properties of the sea ice in the Southern Ocean: "I reassess the assumption that absolute values of sea ice concentration contain information on the sea-ice type in the Antarctic…". Throughout the MS, the author makes reference to waves, free drift sea ice, ice types, dynamical processes, "sea-ice textures", etc, which, to be sure, might not co-vary with sea ice concentration sensed via PM and play a key physical role in Antarctic sea ice evolution. Yet the author provides no supporting information that (a) indeed, the 15-80% threshold does not co-vary with these core sea ice physical properties, or that (b) a \sigma threshold is better, or is related to "ice type" at all.

***Answer***: *The Introduction has been substantially extended to clarify why the threshold-based methos is inadequate. This has been done by describing insights from recent cruises in the Atlantic sector of the Antarctic MIZ (L93-97) and by better clarifying the concept of ice type (L54-67). To help the reader, I have included a series of pictures (new Figure 1) that better illustrate the concept.*

**RC1:** For example, this statement in the discussion: "the proposed analysis will map relative differences between ice types, even if the specific ice type cannot be classified". But how is this true? But what, other than anomalous variability in reported SIC, is actually being measured by this metric? Why does this have anything to do with ice type, and what is the author actually referring to here by "ice type"?

***Answer***: *I acknowledge that I overlooked these concepts, by assuming that they had common meaning. I have indeed removed the sentence and rephrased the conceptual description. In the revised Introduction (L54-67) I have provided more context on how sea ice is described in direct observations and how these features of sea-ice heterogeneity do not co-vary with SIC, although they present features that are typical of MIZ. I added references to the Expert Group on Antarctic Sea-ice Processes and Climate (ASPeCt), as well as the WMO codes to better characterize the ice type. Ice type is indeed an ambiguous term, which is used differently in different contexts and not necessarily linked to thickness and/or concentration. This is now clarified in the introduction, especially how I finally added additional references on the role of frazil ice and how heterogenous Southern Ocean sea ice is (L55-57).*

**RC1**: The author does not provide a physical basis for *how* the MIZ should be defined, anyways, using different terminology at different points throughout before settling on (L281) "variability". Their variability is by construction the anomalous temporal variability of PM-SIC retrievals.

But what the author also emphasizes, as tends to be the case in the literature, is that the MIZ is characterized visually by horizontal variability, i.e. in terms of floe-to-floe heterogeneity, not necessarily temporal variability. Why one should be interchanged with the other is not clear.

*Answer: the revised manuscript contains a restructured Introduction that dedicates a specific section to the concept of variability (Sec. 1.2). With variability I refer to the daily change in SIC over a monthly scale in a climatological sense, and I have further compared spatial and temporal variability in Sec. 3.2. The difference between the temporal variability expressed by this index and the spatial variability has been analysed by comparing with the NOAA/NSIDC CDR derived variable stdev_of_cdr_seaice_conc, which computes the spatial standard deviation of the box of 9 pixels surrounding each pixel. This measure considers the uncertainty of a SIC value based on the variability in the adjacent pixels. I used the monthly average of the latter, and I assumed that the indicator is a valid measure of temporal variability indicative of MIZ conditions when the ratio with the spatial variability is smaller than 1. The ergodic hypothesis assumes that they are interchangeable, but my analysis indicates that temporal variability is larger in the MIZ. A new figure 5 has been added, to show the distribution of this ratio for two selected months, and the full 12 months are included in the supplementary material.*

**RC1:** The evidence supporting the use of this new definition is in part that all four products agree on a climatological seasonal cycle of MIZ extent. The NOAA/NSIDC CDR product used here is simply the maximum value of the NT/BT algorithms (https://doi.org/10.7265/efmz-2t65). Thus the apparent spread in algorithms presented in Fig 5a is in part artificial as NT/BT largely agree, and the CDR product must be smaller than both by definition and should not be compared. As for why the OSI-SAF product produces a more wide distribution of SICs, this has its own substantial literature (e.g. Kern 2019/2020). These algorithms also agree on other metrics too, like SIE. So a global metric with agreement is not altogether all that motivating - there are ways that we know these algorithms all agree, and it may be that the metric you obtain is covariant with one of those. Still, figuring out whether the agreement is "real" requires some further work.

First, it is not necessarily clear they are agreeing for the right reasons: it would be useful to check the marginal ice zone fraction (Horvat, 2021) in concert with the MIZ extent (Rolph et al 2020), as this illustrates whether this agreement is consistent with the same sea ice coverage in general.

*Answer: I have grouped these comments together since they all pertain to the quality of the products and how the proposed indicator reduces the spread in measuring the MIZ extent.*

*I am not entirely sure what the reviewer means by saying that the spread obtained using the SIC threshold is artificial. It has been previously observed (Stroeve et al., 2016) and it comes from applying the same methodology to products that represent the same physical feature. The CDR product is slightly more complex than the maximum value between the BT and NT algorithms especially at the sea-ice edge. To my knowledge, it is a product meant to be an improvement on the individual algorithms. The rationale behind this choice is that PM algorithms tend to underestimate concentration during the summer melt season (Meier et al., 2014). Since greater underestimation is typical in the BT algorithm, then the CDR implements a 10% cutoff of the BT field and then maximises the values between the two. This means that all values lower than 10% from the BT product are not included in the CDR. I do not agree with not comparing the CDR against the individual products, because this choice (driven by considerations on the summer ice conditions) does have an impact on the MIZ estimation. I have added all these concepts in the revised manuscript in L146-154.*

*There is evidence that estimates of MIZ extent from BT and NT do not agree (Stroeve et al. 2016). It is true that they agree on the overall SIE, but the aim of this work is to analyse the MIZ features in*

*Antarctic sea ice. I think the arguments raised by the reviewer are partly reinforcing the conclusions I draw in this work. Despite the known limitations of each product, they all retrieve a similar measure of "variability" in sea ice, which translates into a similar estimate of the climatological MIZ extent. There is more agreement in the seasonality than found with the SIC threshold because the use of anomalies removes some of the biases of the various algorithms. This is to me an indication that there is an underlying physical meaning that goes beyond the technical limitations of each algorithm.*

*This does not mean that the extent obtained through the \sigma is a better estimate of the MIZ. I clearly did not explain properly that the proposed indicator is not meant to substitute the estimates of SIE in the MIZ, because it is not directly comparable with the standard pack ice SIE, or other measures as the MIZ fraction proposed by Horvat (2021). All these concepts have been better explained in this revision, with a more structured Introduction and a revised Discussion section. As explained in L487 onwards, my indicator gives additional information to what Horvat is proposing and would likely help to further assess climate models. I used the binary mask to provide evidence of a much more extended region of variable sea ice that presents conditions more akin to the MIZ. I understand that this is misleading since I used the area as a simpler measure to relate it to the MIZ extent computation. This is clearly shown in Fig. 11, in which the number of pixels affected by higher variance of SIC is larger than the number of pixels that would be classified as sea-ice covered based on the SIE criterion. This has also been flagged by the other reviewer as a part that needs improvement, and it has been further explained in the revised manuscript.*

**RC1:** As the author indicates the use of \sigma can give rise to broaden extents, is it possible that this is covariant with larger \sigma-MIZs? Additionally looking at the spatial coherence of the MIZ definition between different products will also indicate if the \sigma value is the same locationally, or if the definitions agree only when integrated globally.

**Answer:** *I thank the reviewer for this comment. I added a spatial analysis to Sec. 3.2 and a new Fig. 8, which clearly indicate that regions with different SIE and MIZ SIE can have the same value of "intensity of variability". There is a positive linear correlation between total SIE and the MIZ extent, but this is not found in the mean value of the \sigma from the region.*

**RC1:** Further, the author clearly notes that two processes can give rise to high values of \sigma: broad-scale thermodynamic processes that cause the ice edge to retreat/expand, or pixel-scale variability (perhaps caused by storms, though this is not spelled out in detail). There is no exploration of which actually drives this change, but it is sorely needed: a physical driver over \sigma values should be foundational to its definition.

**Answer**: *It is explicitly indicated in the manuscript that this analysis addresses both the seasonal advance/retreat of sea ice and the local variability induced by atmospheric forcing. I am indeed referring to extratropical cyclones, and I have made this statement more explicit in the Introduction (L125-126), in the Methods and in the Discussion (L460-477).*

*As mentioned in one of the previous answers, the current definition of the MIZ is not based on specific physical drivers, and the proposed indicator is still based on SIC data from space. Thermodynamic and dynamic processes both contribute to changes in the temperature brightness, which is the only combined proxy we have. Only in situ data from a large-scale observational system that combines drift and thermodynamic fluxes will produce the proper database to separate these*

*components. These experiments have been done in the Arctic, but not yet in the Antarctic. I agree with the reviewer that this is much needed, but I argue that it would not still be possible based on the existing data from Antarctic sea ice.*

*It is instead possible to separate the role played by synoptic scales from the variability associated to advance/retreat. This is currently the work of a PhD student I am supervising, who published an initial analysis on the association between atmospheric anomalies and sea ice (Hepworth et al., 2022). She is applying a similar methodology but focusing on the 5-7 days scale of polar cyclones. I have added to the revised discussion that a follow up work is currently investigating the two different drivers.*

**RC1**: As mentioned, one very important thing we do know is that all PM-SIC algorithms largely agree on Antarctic sea ice area and extent - so it is possible they also should have similar retreat/extent patterns of the sea ice edge. If this is the leading cause of elevated \sigma values, then the algorithms would agree - \sigma values are simply reflecting a synoptic change which could equally well be observed in the SIC values alone. It might be easy to check this, too - if all monthly values are declining or increasing, then the variability being measured is expansion or retreat of the ice edge, and not intra-monthly heterogeneity in the sea ice.

**Answer**: *If I interpret this comment correctly, it implies that the agreement presented in Fig. 5 is indicative of this method being capable of capturing the seasonal advancement/retreat in a more consistent way. If that would be observable in the absolute value of SIC alone, then the threshold-based estimates would agree. My argument is that the use of the threshold inevitably restricts the extent of the MIZ and its north-south progression because certain regions of sea ice with SIC > 80% are not accounted for, while they have been observed to be classified as MIZ. This is now made clearer with a specific example from an Antarctic sea-ice drifter and a new figure (L300-313, Fig. 6).*

*The proposed indicator is indeed meant to capture the seasonal progression of the MIZ across the Southern Ocean as well as the intra-monthly heterogeneity. As indicated in the previous answer, I realise that this was however only explained in the method section, and it has been expanded in the revised version.*

**RC1:** I could, for example, propose a wholly different metric: what if you produced daily maps of the SIC-threshold MIZ (i.e. identified points with 15-80% SIC every day), and averaged this binary indicator over each month instead of defining the threshold on the monthly climatology? How different would this look from the "variability" metric, e.g. in Fig.s 3-4? Why is this metric any better or worse?

**Answer**: *I agree that this method would add an intensity to the binary mask provided by the SIC threshold, thus making it more similar to the proposed indicator. However, binary indicators based on the 15-80 threshold would still not detect changes when sea ice is above 80%, a condition often found in the MIZ. I do see the point made by the reviewer and I have included alternative definitions in the discussion (L503-511). I am glad that this manuscript is leading to a further search for alternative ways of determining the MIZ state. Such an indicator could be useful to detect the type of*

*seasonal progression. For instance, if we assume a linear increase in sea ice over a month from 0% to 100%, the average of this binary indicator will tend to 0.65. Other fractions may be indicative of different types of seasonal growth conditions.*

**RC1:** Finally, there is no discussion of the influence on retrieval uncertainty on \sigma results, and there ought to be. Such errors directly impact th11g e variability measure but will not impact the SIC thresholding (unless occurring at 15\% or 80\% SIC), which is why extent and the MIZ are designed in the way they are. There can be immense variability day-to-day, and errors for non-compact ice can be high. Without a formal assessment of the impact of measurement uncertainty, it is not possible to asses whether there is any true variability being measured. A particular problem raised in the PM observational literature is the "truncation" of SIC estimates (see Kern et al 2019) - most algorithms frequently can return SIC > 1, and then set SIC > 1 to 1. But this can bias the statistics of metrics like \sigma, and shouldn't because it reflect a real "observation". The OSI-450 product is a good choice here because it actually reports the true SIC estimate, which can be used in your assessment of the variability and extent (this field is raw_ice_conc_values in OSI-450 output).

**Answer**: *The reviewer is entirely right on this part, which has been investigated but not properly included. I agree that this uncertainty may have a higher impact on the indicator than on the threshold method. In the revised version I have included a comparison with the spatial standard deviation (a measure of uncertainty in the CDR), to show that the \sigma signal is higher than this uncertainty in the MIZ (Sec. 3.2, L277-280). The mean climatological values for the months of December and August are shown in Fig. 5, chosen as examples of austral summer and winter months before the months of minimum and maximum extent. The ratio between the spatial uncertainty and \sigma is order 10^{-1} in the MIZ.*

*I have also added a paragraph in Section 4.2 to discuss how the use of "filters" like capping, ocean pixels and time/space interpolation may impact the application of the method, and in general the derivation of variability measures from data sets that have been intended as diagnostics for the essential climate variables. This is now necessary since the version 4 of the dataset I used in the present manuscript implements these filters by default. I have kept some of the figures from the previous analysis in the supplementary material to support the discussion.*

RC1: Finally, the discussion circles around the meaning of variability without doing any direct comparison to other observations. I have mentioned the many asides to MIZ physics and ice types, which is not reflected in the product itself (and is readily admitted by the author, see L270), nor supported in the analysis. These should be a major part of what makes this definition useful, but they do not support its inclusion.

**Answer**: *This improvement has been made in the revised version (new Section 3.2). Please see the answer to the first suggestion below for the details.*

**RC1:** Suggestions

I make two overarching suggestions which I hope would render this article a significant contribution to the sea-ice literature.

First, the author should relate the new definition to some physical properties of the sea ice cover relevant for those who might be interested in this definition. It is true that the current MIZ definition was simply defined operationally. But an alternate definition should have additional reasons for its suggestion. This would require the use of alternate data, i.e. a case study in a particular region with imagery, or similar, to give evidence that high \sigma regions are indeed compatible with a physical definition of the MIZ. Datasets on sea ice age, floe size, waves, surface roughness, etc, all do exist and could be used to further this effort.

*Answer: I thank the reviewer for this suggestion, which has been implemented in the revised manuscript. I acknowledge the succinct description that I provided in the introduction about the need to ground the alternate definition on physical conditions observed in the MIZ, and I have substantially expanded the Introduction with a more structured description of the current definitions and the need for a revised complementary diagnostic. In the Discussion (L448), I hint at the importance to use a multivariate approach for the MIZ definition that combines mean and variance.*

*I agree that descriptive MIZ features can be obtained from literature, although these datasets are unlikely to be comprehensive enough in a spatial and temporal sense. I have selected a few examples in the new Section 3.2, in which I do a critical assessment of the indicator and a direct comparison with the threshold-based MIZ definition. I am not aware of any dataset or product related to sea ice age in Antarctica. To the best of my knowledge, the NSIDC age product (: https://doi.org/10.5067/UTAV7490FEPB., Tschudi et al, 2020) is only available for the Arctic.*

*The assessment has been done considering the climatological intent of determining regions of higher variability that is at the basis of this approach. A comparison with instantaneous observations (e.g. SAR images) or short-term cruises is not applicable for a climatological indicator, and this has been explained as follows (L290-298):*

*"I argue that this question cannot be adequately answered for two main reasons: 1) the use of a threshold-based MIZ has not been objectively assessed in the literature but merely applied operationally, which poses a considerable challenge when proposing any alternative indicator; 2) there are no ancillary observational datasets (at least not derived from passive microwave measurements) that would allow an independent assessment of any metrics. MIZ diagnostics are usually applied in climatological or integrated analyses (for shorter times and specific regions, SIC is the variable of preference), and as such it is difficult to assess them against local ship observations or SAR images. However, these points should not dissuade us from comparing with data that have sufficient time coverage, as for instance buoy data lasting longer than a month, or comparing the different metrics without a benchmark, as typically done in model intercomparisons projects."*

**RC1:** Second, the author should separate the aforementioned sources of variability into that due to ice-edge retreat, real inter-monthly variability in ice conditions, and PM uncertainty. This is necessary to know whether \sigma actually contains useful information or is just reflecting uncertainty at the ice edge. Perhaps it is! That might be a useful back-door way of observing the MIZ, but without knowing it is impossible to do more than speculate.

*Answer*: As explained in the answers to the comments above, the aim of this paper is to jointly analyse the two main sources of variability in Antarctic MIZ: seasonal advance/retreat and subseasonal changes driven by synoptic atmospheric features. This point has been made more explicit in the revised version. I acknowledge the importance of including an analysis of PM uncertainty, which has been left out from the current manuscript, which has been included as requested (new section 3.2).

I would however prefer to leave the discrimination of the source of variability to a further work, since this is the topic of a PhD thesis that started 2 years ago, and the results will be submitted within this year. An initial analysis towards the role played by polar cyclones has been published by Hepworth et al (2022). This paper has now been included in the discussion (L460-467).

This paper presents a new method to map the MIZ that was originally mapped by using the 15-85% of sea ice concentration (SIC) from passive microwave remote sensing data. Different algorithms in deriving the SIC would give very different MIZ (extent). But the new method that using the standard deviation of daily SIC anomalies (on monthly basis) gives consistent MIZ (extent) based on the SIC derived from different algorithms/datasets (Figure 5). Therefore, the paper concludes that this new method is a better method as compared with the 15-85% method, although without thorough evaluation to see if this is indeed the best MIZ (extent). I would think this is a new method and deserves further investigation and I encourage the author to do so.

**RC2**: I would think the very first addition to confirm the potential effectiveness of the method is to apply this method to the Arctic sea ice. If the same conclusion is achieved, I would think it might be effective. Another way to evaluate the method is to compare the MIZ derived from high resolution imagery, especially for those areas and periods (for example, later spring/summer) with the highest disparity among the new method and existing methods.

***Answer***: *The literature on the MIZ definition in the Arctic using the operational threshold is much more extended than in the Antarctic. This work addresses the limitations of the method when applied to Antarctic sea ice, but I acknowledge that I have not provided enough supporting literature to indicate the need for a new definition. This has been pointed out by Reviewer 1, and it has been addressed in the revised version of the manuscript with a much more structured Introduction. The extent of the MIZ is more limited in Arctic sea ice, although recent literature is indicating that there are shortcomings in the simulation of the MIZ fraction in climate models (Horvat, 2021). I would argue that a complete analysis of this indicator in the Arctic would be beyond the scope of this work, but I agree that some indications are useful and have been included in the revision (L240-245). I have added a new panel in Fig. 3, which compares the median and spatial distribution of the \sigma indicator between the two hemispheres. The empirical distribution of the median is also different from the Antarctic, which as been described as follows in the text: "The number of pixels with low variability is larger, as known to be in the Arctic due to the presence of multi-year ice, and the second peak is lower and barely visible. There is instead a plateau of points that show median values of the indicator between 0.05 and 0.17, and a clear threshold as the one seen in the Antarctic is less distinguishable."*

*I have added Section 3.2 to critically assess the proposed diagnostic, as also requested by the other reviewer. Descriptive MIZ features can be obtained from literature, although these datasets are unlikely to be comprehensive enough in a spatial and temporal sense. A comparison with instantaneous observations (e.g. SAR images) is not appropriate, given the climatological context of the indicator. I have added an analysis of a long-term drifters (Womack et al., 2022), and expanded the comparison with the threshold-based indicator to show the power in detecting larger scale features during the extreme event in November 2016.*

**RC2**: Second, as indicated in the introduction, that SIC based MIZ identification is more reliable in the wintertime in southern oceans, I would agree your method seems achieve similar results (make sure this is correct), but for summer time, especially Nov, Dec, your results show too much high extent (Figure 5), similar or even larger, as compared with these from the 15-85% method that already said they are not accurate. Since overall, the Nov and Dec ice extents are smaller than the Sep/Oct, I

would say the MIZ (extent) should be smaller than the Sep/Oct MIZ (extent). I know your statistic-based MIZ include those of the polynyas, not sure if these should be excluded? MIZ-like statistics can also found in the interior of the pack ice, should these zones also included as MIZ?

*Answer: The reviewer is correct that Fig. 5 (of the former manuscript, now Fig. 10) may suggest some misleading conclusions. I acknowledge some ambiguity in the use of this method to compare with the traditional SIE. This method is originally designed to diagnose a more appropriate monthly climatology that would improve the current operational definition. Specifically, it is meant to address whether a region of the seasonally covered ocean is characterised by relatively high temporal variations in SIC, a metric that cannot be obtained by the 15-80% threshold.  A new Section 4.2 in the Discussion has been included to better explain the outcomes of the analysis and comparison with the 15-80% threshold. At L503-511, I give examples of alternative indicators, and explain that I do not mean to say that the threshold-based estimates are not accurate, but that there are regions of the ice-covered ocean that present physical characteristics similar to the MIZ even when the concentration is above 80%. This includes areas within the pack ice and areas of polynyas. My analysis is therefore more oriented towards the estimation of variability due to heterogeneous ice conditions, independently of where they are located. For this reason I have changed the title in the revised manuscript to indicate that the work first addresses the variability and how this can affect the description of the MIZ*

*.*

**RC2**: In figure 6, your MIZ (yellow) for the December seems way to bigger and this makes me doubt your method for the later spring (Nov/Dec). maybe you need to use a larger threshold value for this period? Instead of 0.1, maybe 0.15 for this case? In Figure 7, the MIZ (extent) is larger than the SIE in five months, needing good explanation. To me the MIZ (extent) from the NOAA ORD data seems more reasonable (all smaller than the SIE) (Figure 7). In line 227-228, you mentioned "climatological MIZ extent shown in Fig 5 is an underestimation of sea ice area", but then in line 232, you said that "MIZ extent presented in this work exceeds the total SIE". Some confusions here needing explanation.

*Answer: This comment is related to the point above. I have now explained, both in the revised Introduction and in the Discussion sections, that this method is complementary to the use of the MIZ extent when compared with SIE (also known as the MIZ fraction; see Horvat, 2021, reference included). It provides a different information, and for this reason I'm arguing in the Discussion that we need a multivariate definition to properly capture all the features. Former Fig. 7 (now Fig. 11) has been retained to illustrate that there is a fundamental computational difference between the climatological averaging of the monthly extents shown in former Fig. 5b (now Fig. 9), in which a monthly mask is multiplied by the pixel area then integrated and averaged, and the mask based on the climatological monthly standard deviation of the daily anomalies. This is because the average of the standard deviations computed from sub-samples of a population is different from the standard deviation of the whole population.*

*To address these issues, in the revised manuscript, I have:*

1) *added the new section 3.2 that better characterises the uncertainties by comparing the spatial standard deviation with the proposed indicator. This analysis revealed that the variability signal obtained in the Antarctic MIZ is robust to be used for its characterisation in all seasons, including summer.*
2) *Offered a critical assessment of the caveats of this indicator in the new section 4.2*

**RC2**: third, in the figure 5, I believe this is the 30-40 year averages, right? can you show a at least a sub-set of the those in each year? say 2008, 2009, 2010, 2011...; so make sure those differences also seen in yearly curves, not just an effect of average of 30 years or 40 years...

*Answer: yes, this is the climatology. In the revised manuscript I have presented specific years in Supplementary Fig. S8. They have been commented at L372-375. The agreement between products is not a result of the averaging.*

**RC2**: fourth, your taking of 0.1 for the σ value seems random, why not 0.12, 0.15, 0.17, or 0.2...? should this number the same for the Arctic sea ice?

*Answer: this number is obtained from the analysis of the median distribution shown in Fig. 3 (Fig. 2b of the previous manuscript). It is the representative value of the trough of the distribution. The results are not sensitive to 20% variations around this value, and this has now been indicated in the revised manuscript. Fig. 3b now shows the distribution for the Arctic, that clearly shows that such threshold is not present, as expected given the much lower variability of the MIZ.*